

# How Rossby wave breaking modulates the water cycle in the North Atlantic trade wind region

Franziska Aemisegger[1], Raphaela Vogel[2], Pascal Graf[1], Fabienne Dahinden[1], Leonie Villiger[1], Friedhelm Jansen[3], Sandrine Bony[2], Bjorn Stevens[3], and Heini Wernli[1]

[1]Institute for Atmospheric and Climate Science, ETH Zurich, Zurich, 8092, Switzerland
[2]LMD/IPSL, Sorbonne University, CNRS, Paris, France
[3]Max Planck Institute for Meteorology, Hamburg, Germany

*Correspondence to*: Franziska Aemisegger (franziska.aemisegger@env.ethz.ch)

**Abstract.** The interaction between low-level tropical clouds and the large-scale circulation is a key feedback element in our climate system, but our understanding of it is still fragmentary. In this paper, the role of upper-level extratropical dynamics for the development of contrasting shallow cumulus cloud patterns in the western North Atlantic trade wind region is investigated. Stable water isotopes are used as tracers for the origin of air parcels arriving in the sub-cloud layer above Barbados, measured continuously in water vapour at the Barbados Cloud Observatory during a 24-day measurement campaign (isoTrades, 25 January to 17 February 2018). This data is combined with a detailed air parcel back-trajectory analysis using hourly ERA5 reanalyses of the European Centre for Medium Range Weather Forecasts. A climatological investigation of the 10-day air parcel history for January and February in the recent decade shows that 55% of the air parcels arriving in the sub-cloud layer have spent at least one day in the extratropics (north of 35°N) before arriving in the eastern Caribbean at about 13°N. In 2018, this share of air parcels with extratropical origin was anomalously large with 88%. In two detailed case studies during the
campaign, two flow regimes with distinct isotope signatures transporting extratropical air into the Caribbean are investigated. In both regimes, the air parcels descend from the lower part of the midlatitude jet stream towards the equator, at the eastern edge of subtropical anticyclones, in the context of Rossby wave breaking events. The zonal location of the wave breaking, and the surface anticyclone, determines the dominant transport regime. The first regime represents the "typical" trade wind situation with easterly winds bringing moist air from the eastern North Atlantic into the Caribbean, in a deep layer from the
surface up to ~600 hPa. The moisture source of the sub-cloud layer water vapour is located on average 2000 km upstream of Barbados. In this regime, Rossby wave breaking and the descent of air from the extratropics occurs in the eastern North Atlantic, at about 33°W. The second regime is associated with air parcels descending slantwise by on average 300 hPa (6 d)$^{-1}$ directly from the northeast, i.e., at about 50°W. These originally dry airstreams experience a more rapid moistening than typical trade wind air parcels when interacting with the subtropical oceanic boundary layer, with moisture sources being located on
average 1350 km upstream to the northeast of Barbados. The descent of dry air in the second regime can be steered towards the Caribbean by the interplay of a persistent upper-level cutoff low over the central North Atlantic (about 45°W) and the associated surface cyclone underneath. The zonal location of Rossby wave breaking, and consequently, the pathway of





extratropical air towards the Caribbean, is shown to be relevant for the sub-cloud layer humidity and shallow cumulus cloud cover properties of the North Atlantic winter trades. Overall, this study highlights the importance of extratropical dynamical
processes for the tropical water cycle and reveals that these processes lead to a substantial modulation of stable water isotope signals in the near-surface humidity.

## 1 Introduction

Understanding and correctly predicting the patterns of shallow cloudiness over the trade-wind dominated tropical oceans is one of the current big challenges of climate research (Bony and Stevens, 2012). The interaction of these low-level clouds with
their large-scale environment and their impact on Earth's radiation budget are key feedback elements in our climate system (Bony et al., 2015). A large part of the uncertainty in current climate models is thought to be related to the models' representation of the strength of vertical mixing of water vapour in the lowest kilometres of the atmosphere, in particular between the boundary layer and the free troposphere (Sherwood et al., 2014). The balance between convective drying by mixing in of free tropospheric dry air and turbulent moistening of the boundary layer by ocean evaporation under different
large-scale forcing situations is an important element in the process chain of shallow cumulus cloud formation. Furthermore, evaporation of falling rain drops below low-level clouds into unsaturated downdrafts can trigger density currents spreading out at the surface and leading to the formation of convective cold pools. The gust fronts of propagating cold pools force environmental air to rise and can thereby trigger new convection, clouds and precipitation (Purdom, 1976; Weaver and Nelson, 1982; Zuidema et al., 2012; Torri et al., 2015). The three main components of the boundary layer moisture budget (Risi et al.,
2019) namely (1) ocean evaporation, (2) convective drying, and (3) moistening by hydrometeor evaporation carry a distinct signature in their stable water isotope composition (Aemisegger et al., 2015, Benetti et al., 2015, Sodemann et al., 2017, Aemisegger and Sjolte, 2018, Graf et al., 2019).

Given the strong contrasts in the isotope signature of the different sources mentioned above, isotope meteorology could be used as an observation-based measure to evaluate the influence of the large-scale circulation on shallow cumulus cloud
patterns. The recent studies by Scholl and Murphy (2014a) and Torri et al. (2017) took first steps towards using stable water isotopes to link convection in the tropics to the large-scale circulation. Due to the scarcity of available data with high precision and high time resolution in the tropics, Torri et al. (2017) relied on monthly precipitation isotope information from the Global Network of Isotopes in Precipitation stations of the International Atomic Energy Agency (IAEA) and Scholl and Murphy (2014a) used weekly collected precipitation and a few cloud water samples (Scholl et al., 2009, Scholl et al., 2014b). However,
at these aggregated timescales a large part of the interesting dynamics of convective activity that is responsible for cloud organisation is lost. Studies focussing on short timescale variability are thus in great need.

In this paper, the hourly stable water isotope signature of the tropical trade wind sub-cloud layer measured at the Barbados Cloud Observatory (BCO, black cross in Fig. 1, Stevens et al., 2016) during a 24-day campaign in January-February 2018 (isoTrades) is used in combination with a detailed air parcel back-trajectory analysis based on ERA5 reanalysis data from the



European Centre for Medium Range Weather Forecasts (ECMWF). The measured stable water isotope signals are used as tracers for moist atmospheric processes informing about moisture source and transport characteristics. Thereby, the role of the large-scale flow for lower tropospheric humidity and the shallow cumulus cloud patterns is investigated in the western North Atlantic trade wind region.

The observed patterns of clouds in the trade wind region resulting from the organisation of shallow convection is embedded in and interacts with the descending large-scale flow. The latter is typically associated with the circulations of the tropical Hadley and the midlatitude Ferrel cells (Hadley, 1735; Ferrel, 1856). Dry air masses originating from the tropical deep convection outflows, the extratropical jet stream region or from the subtropical mid troposphere descend towards the surface, thereby generally stabilising and dehydrating the atmosphere above the cloud layer (Yoneyama and Parsons, 1999; Cau et al., 2005). The subsidence pathways and progressive moistening of theses airstreams by surface fluxes, turbulent mixing and convection (Lee et al., 2011; Brown et al., 2013, Lee et al., 2019) are key preconditioning factors that shape the atmospheric environment in which shallow convection develops. Note that in this paper we use "subsidence" and "descent" as synonyms when referring to slantwise descending motion of air parcels.

A dynamical phenomenon with a particularly strong impact on the vertical extent of shallow cumulus clouds are so-called dry air tongues (Mapes and Zuidema, 1996) or dry intrusions (Browning, 1993, Raveh-Rubin, 2017). These dry intrusions have been shown to originate from the midlatitudes (Yoneyama and Parsons, 1999) using soundings and global atmospheric analysis datasets from the Coupled Ocean–Atmosphere Response Experiment of the Tropical Ocean and Global Atmosphere programme (TOGA COARE; Webster and Lukas, 1992). The impact of extratropical dry intrusions on the occurrence and duration of convective systems in the West African monsoon was studied in detail in Roca et al. (2005). In a few cases, the strongly descending air in the dry intrusion originates from the lower stratosphere (e.g., Wernli, 1997; Raveh-Rubin, 2017) and these events are also referred to as stratospheric intrusions. The isotope composition of such a stratospheric intrusion has been measured on the Chajnantor Plateau in northern Chile in a study by Galewsky and Samuels-Crow (2014). Over the North Atlantic, the moistening of dry intrusion air parcels when reaching the marine boundary layer is expected to be more important than on the Chajnantor Plateau. Enhanced ocean evaporation due to dry and cold air advection may lead to a dominant impact on the stable water isotope signals in the marine boundary layer (Aemisegger and Sjolte, 2018). Therefore, the response of the tropical low-level mesoscale circulation to the extratropical disturbance during such events of dry intrusions is essential for understanding the variability of low clouds in the tropics, and eventually also how they respond to climate change.

The overall aim of this paper is to identify the moisture transport pathways of air parcels arriving in the sub-cloud layer in Barbados in winter (January-February) based on their isotopic signature and back-trajectory analysis. We therefore organised the paper around four attendant objectives, which are 1) to quantify the climatological importance of transport pathways from the extratropics, 2) to provide observational evidence for the occurrence of extratropical dry intrusions in Barbados using stable water isotope measurements as tracers for moisture source and transport processes, 3) to quantify the impact of extratropical dry intrusions on the sub-cloud layer properties, and 4) to characterise the midlatitude dynamical flow context in which air parcels with extratropical origin start their subsidence pathway into the Caribbean. The remainder of the paper is structured as





follows: in Section 2, a description of the used Lagrangian diagnostics based on ERA5 reanalyses is provided as well as a
summary of the measurement setup and postprocessing of the isotope data; in Section 3, the results addressing the four
objectives formulated above are presented and discussed; and a concluding summary is given in Section 4.

## 2 Data and methods

### 2.1 Large-scale flow characteristics and Lagrangian diagnostics

The ERA5 reanalysis dataset (Copernicus Climate Change Service, 2017; Hersbach et al., 2019; Hersbach et al., 2020) from
the ECMWF was used to describe the three-dimensional large-scale flow situation during isoTrades in 2018, to calculate
backward trajectories for the extended period January-February 2009 to 2018, and to derive several Lagrangian transport
diagnostics as further described below. In this study, the hourly ERA5 reanalysis data is interpolated to a regular horizontal
grid with 0.5° spacing.

First evaluations of the difference between ECMWF's predecessor reanalysis product ERA-Interim and ERA5 show that
spatial transport deviations between the two datasets can be up to an order of magnitude larger than those caused by
parameterised diffusion and subgrid-scale wind fluctuations after one day (Hoffmann et al., 2019). Hoffmann et al. (2019)
found differences of up to 30% in specific humidity along back-trajectories calculated with the two datasets after one day.
They showed that in addition to the improved spatial resolution in ERA5, changes in the forecast model, available observations
and in the data assimilation system all play a role in the observed deviations in transport and thermodynamic conditions along
back-trajectories between the two datasets. Here the added value of hourly data in ERA5 (ERA-Interim is only available 6-
hourly) and the increased spatial resolution is the primary reason for using the new reanalysis dataset.

### 2.1.1 Trajectory calculation

Ten-day backward trajectories were calculated with the Lagrangian Analysis Tool (LAGRANTO, Wernli and Davies; 1997,
Sprenger and Wernli, 2015) based on the three-dimensional wind fields from ERA5. The trajectories were started hourly above
the geographical position of the BCO (13.16°N, 59.43°W, Fig. 1) as well as 4 other points displaced zonally and meridionally
by 0.5° from the BCO to account for the uncertainty of trajectory calculations with the resolved wind. The starting points were
vertically stacked every 7.5 hPa between 1000 hPa and 940 hPa for January-February in the period 2009-2018 and up to
200 hPa in 2018 to be able to also look at the transport at higher levels. For the case studies during isoTrades, we separately
considered two layers with trajectory arrival points: the sub-cloud layer ($p \geq 940$ hPa) and the cloud layer ($940 > p \geq 700$ hPa).
Even though the vertical extent of the sub-cloud and cloud layers is variable in time and depends on the strength of the shallow
convective activity, we choose to use fixed vertical layers. The reason for this simplifying choice is that the temporal variability
is not large, and the precision of our subsequent calculations would not benefit from accounting for this variability. We defined
the cloud base height at 940 hPa (~630 m) and the cloud top height at 700 hPa (~3 km). The chosen cloud base level is in close





agreement with the estimate of Nujiens et al. (2014), who found a cloud base level from the BCO humidity and temperature
measurements at 700±150 m. The top of the cloud layer is chosen slightly above the mean level of the trade wind inversion
(2 km). With this setup, the sub-cloud layer air consists of 40 air parcels per hourly time step, and the cloud layer of 150 air
parcels. For the majority of the analyses conducted in this paper, in particular for the combination with the near-surface stable
water isotope measurements, the trajectories arriving in the sub-cloud layer are used. Note that the spread of the trajectories
calculated from different starting points in a given vertical layer allows us to take into account the uncertainty of the trajectory
calculation and to capture the effect of mixing of air parcels from different origin.

### 2.1.2 Trajectory-based diagnostics

The following four types of trajectory-based diagnostics are used in this paper and introduced here:

1) Air parcel residence times in different spherical caps:

The residence time ($\tau_\lambda$) of air parcels in different Northern Hemisphere caps (north of $\lambda$=23.5°N, 30°N, 35°N, 40°N, 50°N,
60°N), is calculated based on the position of the air parcels within 10 days before their arrival at the BCO. Thus, e.g. the
residence time north of 23.5°N is referred to as $\tau_{23.5}$. The residence time in the extratropics is defined as $\tau_{35}$. The
exceedance probabilities of $\tau$, i.e. the occurrence frequency of air parcels with $\tau \geq x$ days in a given latitudinal band
provides a climatological measure of the origin of air parcels arriving in Barbados. The exceedance probability of $\tau$ for
the range 0 to 10 days in the defined spherical caps is calculated for a climatology of the recent 10 years as well as
separately for 2018, in order to place the year of the isoTrades campaign in a climatological context. The result of this
analysis is shown in Fig. 3 and discussed in Section 3.1.

2) Three flow regimes:

Air parcels arriving in the sub-cloud layer in Barbados are classified into three regimes: the extratropical dry intrusion
regime, the extratropical trade wind regime and the tropical regime (see schematics in Fig. 2). For the moistening of the
sub-cloud layer air parcels, their latitudinal origin and the characteristics of their subsidence pathway is of particular
importance. Therefore, the definition of the three flow regimes used in this study is based on the median residence time
in the extratropics $\tilde{\tau}_{35}$ and the median amplitude of the subsidence ($\widetilde{\Delta p}$) of air parcels. The **tropical flow regime** is
distinguished from the two extratropical flow regimes based on $\tilde{\tau}_{35} < 1$ day. If $\tilde{\tau}_{35} \geq 1$ day, the time step is classified into
one of the extratropical flow regimes, which are differentiated based on $\widetilde{\Delta p}_{4d}$ in the four days prior to arrival of the air
parcels in Barbados, with $\Delta p_{4d} = p_{tBCO} - p_{tBCO-4d}$, where $p$ is the air parcel pressure and $t_{BCO}$ the arrival time in
Barbados. If $\widetilde{\Delta p}_{4d} \geq 100$ hPa (4 d)$^{-1}$ the date is classified into the **extratropical dry intrusion flow regime**. If $\widetilde{\Delta p}_{4d} < 100$
hPa (4 d)$^{-1}$, the date is classified into the **extratropical trade wind flow regime** with sub-cloud layer air originating from
the extratropics but having experienced limited subsidence in the four days prior to arrival in Barbados. A more detailed
justification for the chosen flow regime classification framework can be found in Appendix A.





The occurrence frequencies of the different regimes during the isoTrades campaign and in the climatological period 2009-

2018 are summarised in Table 1 and discussed in Section 3.1. To characterise the subsidence behaviour of the air parcels

arriving in the sub-cloud layer in Barbados in the three defined flow regimes, the hourly median pressure ($\tilde{p}$) and two-day

subsidence rate $\widetilde{\Delta p_{2d}}$ are calculated for 1 to 10 days before arrival in Barbados. Additionally, for the campaign period in

January-February 2018, several flow regime average characteristics are calculated from the ERA5 trajectories and from

the local meteorological and isotope variables measured at the BCO. These values are shown in Tables 2 and 3 and shortly

discussed in Section 3.2. Given the low occurrence frequency of the tropical flow regime in 2018 (6%), this flow regime

is not further analysed in terms of its associated isotope signature and sub-cloud layer properties due to the small sample

size.

3)    Characteristics of maximum slantwise descent:

To investigate the dynamical environment in which the subsidence from the extratropics occurs, the two-day period of

maximum subsidence is identified for each trajectory ($\max(\Delta p_{2d})$). Time, pressure, latitude, longitude, specific and relative

humidity are calculated for the start and end of the period of maximum subsidence for all the trajectories arriving in the

sub-cloud layer and the average for each flow regime is calculated and shown in Table 2.

4)    Moisture sources and moisture uptake characteristics:

The moisture sources of the air parcels arriving in the sub-cloud layer are identified using the method of Sodemann et al.

(2008). In short, this method considers the mass budget of water vapour in an air parcel. Moisture uptakes are registered,

whenever the specific humidity along an air parcel increases. The weight of each uptake depends on its contribution to the

final humidity of the trajectory. If precipitation occurs (i.e. a decrease of specific humidity along the trajectory happens)

after one or several uptakes, the weight of all previous uptakes is reduced proportionally to their respective contribution

to the loss. The moisture sources identified for each trajectory are subsequently weighted by the air parcel's specific

humidity at the arrival in the sub-cloud layer. This method has been used extensively, in particular for the interpretation

of water isotope signals in vapour and precipitation (e.g., Pfahl and Wernli, 2008; Aemisegger et al., 2014; Aemisegger,

2018; Thurnherr et al., 2020). In addition to the moisture source regions, the moisture uptake during the two-day period

of maximum subsidence as well as during the period after maximum subsidence until arrival in Barbados is calculated

and the average for each regime is given in Table 2.

## 2.2 Meteorological data from the Barbados Cloud Observatory

The BCO (https://barbados.mpimet.mpg.de/) was set up on a promontory near the most windward point of the Island of

Barbados at Deebles Point (13.1626°N, 59.4287°W, see Fig. 1) in 2009 (Stevens et al., 2016) and is operated as a cooperative

research project of the Max Planck Institute for Meteorology, the Caribbean Institute for Meteorology and Hydrology and the

Museum of Barbados. Situated on a cliff at 17 m a.s.l., the BCO is directly exposed to the North Atlantic trade winds coming

in from the east or northeast. Previous work has not been able to identify a significant island effect on the measurements.

During December to May, Barbados is exposed to a typical trade wind flow with prevailing low-level easterlies and large-





scale subsidence at upper levels. Shallow cumulus clouds can be observed throughout the year (Nuijens et al., 2014), organised in a variety of mesoscale cloud patterns (Stevens et al., 2020a). Medeiros and Nuijens (2016) showed that observed and

simulated clouds near Barbados are representative for much of the tropical oceans. Barbados therefore represents an ideal study site of shallow cumulus clouds in the trades.

Meteorological data from a Vaisala WXT-520 mounted on a 3 m mast at the BCO was used for calculating mean conditions during the different flow regimes. Cold pools affecting the BCO were identified following the method introduced in Vogel (2017) based on 1 min surface temperature data from the BCO. Radio sounding data from the Grantley Adams Barbados

Airport (15 km from the BCO) were used in the two case studies presented in Section 3.4 to characterise the contrasting lower tropospheric thermodynamic conditions in the two extratropical flow regimes.

## 2.3 Stable water isotope measurements at the Barbados Cloud Observatory

In preparation of the large international field campaign "Elucidating the role of clouds-circulation coupling in climate" (EUREC[4]A) in January-February 2020 (Bony et al., 2017, Stevens et al., 2020b), a 24-day field experiment focusing on stable

water isotope measurements (isoTrades) was carried out at the BCO. The water vapour isotope measurements and event-based precipitation samples from isoTrades served as a basis for the planning of the multi-platform isotope measurements on four ships, two aircrafts, and at the BCO performed during EUREC[4]A-iso by several European and U.S. American teams.

The stable water isotope composition of a water sample is usually quantified by the $\delta$ notation (Craig, 1961a): $\delta\,[\text{‰}] = (R_{\text{sample}}/R_{\text{VSMOW}} - 1) \cdot 1000$, where $R$ is the isotopic ratio of either $H_2^{18}O$ or $^2H^1H^{16}O$ (with $R$ representing the

ratio of the concentration of the heavy molecule to the concentration of $H_2^{16}O$). The $\delta$ notation expresses the relative deviation of the isotopic (molecular) ratios from the internationally accepted primary water isotope standard, that is, the Vienna standard mean ocean water (VSMOW2, IAEA, 2017; with $^2R_{\text{VSMOW2}}=3.1152\times10^{-4}$ and $^{18}R_{\text{VSMOW2}}=2.0052\times10^{-3}$). The second order isotope parameter deuterium excess ($d = \delta^2H - 8 \cdot \delta^{18}O$, Dansgaard, 1964) serves as a tracer for non-equilibrium fractionation (Craig and Gordon, 1965; Pfahl and Wernli, 2008), in particular for events of strong large-scale ocean evaporation

(Aemisegger and Sjolte, 2018).

The high-resolution temporal variability of water isotopes can be measured directly in the gas phase owing to recent advances in laser spectrometric devices that have reached sufficiently high precision (Baer et al., 2002; Kerstel, 2004; Crosson, 2008) and have been widely used in the field (Wei et al., 2019). As a part of isoTrades, a customised fast-response version of a L2130 Picarro cavity ring-down laser spectrometer characterised in detail in Aemisegger et al. (2012) and in the supplement of

Thurnherr et al. (2020) was installed in a temperature-regulated container ($T_{\text{container}} = 24 \pm 2$ °C) at the BCO. The setup is described in Fig. 1b. In short, the inlet is shielded from rainfall and sea spray by a funnel through which a heated inlet line (80°C, Winkler, Germany) guides the ambient air into the container. The inlet line material is PTFE, it has an outer diameter of 12 mm and a length of 9 m with 5.5 m outside, and 3.5 m inside the container (red line in Fig. 1b). The heated inlet line is flushed with the inlet pump (P1, KNF HN022AN.18). The sub-sample of the ambient gas is guided to the cavity ring-down

system through a 30 cm isolated PTFE sample line with an outer diameter of ¼ inch (green lines) by the instrument external



pump (P2, KNF N920AP.29.18). All together the residence time in the system is 6 s with 3 s in the inlet system and 3 s in the instrument. For the normalisation to the international IAEA VSMOW-SLAP scale and for drift correction, the isotope composition of known liquid standards was measured daily for 20 min by the cavity ring-down system using a vaporiser and a standard delivery module from Picarro. The isotope composition of the standards used was (4.77±0.2) ‰, (−11.4±0.2) ‰

and (−34.5±0.1) ‰ for $\delta^{18}O$, and (42.7±0.4) ‰, (−82.1±0.6) ‰ and (−267.8±0.1) ‰ for $\delta^2H$, respectively. Subsamples of the liquid standards were taken on the first and last day of the campaign to correct for a potential drift in the standard's isotope composition, which was determined in the same way and by the same laboratory as the precipitation samples (see below). The vapour isotope data is post-processed and normalised to the VSMOW-VSLAP2 scale (Gonfiantini, 1978, IAEA, 2017) following the procedure described in Aemisegger et al. (2012). The measurement uncertainty based on a conservative estimate

using error propagation is 0.8 ‰, 1.7 ‰ and 1.9 ‰ for $\delta^{18}O$, $\delta^2H$, and $d$, respectively. From the daily calibration runs performed during the campaign a more realistic estimate of the uncertainty can be obtained based on the root mean square deviation of the calibrated data to the reference standard values, which yields 0.3 ‰, 0.7 ‰ and 0.8 ‰ for $\delta^{18}O$, $\delta^2H$, and $d$, respectively. For the combination with the hourly ERA5 backward trajectory analysis, the 1 Hz water vapour isotope data was averaged to hourly data (available online, see Aemisegger and Graf, 2020).

The precipitation samples were collected as soon as possible after a rainfall event using a precipitation sampling system that is especially designed to avoid post-sampling re-evaporation (PALMEX RS1). The same sampling system has also been used by the IAEA in its Global Network for Isotopes in Precipitation. The uncertainty due to small scale variability of rainfall was assessed in Europe in a dedicated study on 10 precipitation events with an array of similar samplers and found to be < 2 ‰ in $\delta^2H$ and < 0.3 ‰ in $\delta^{18}O$ (Fischer et al., 2019). The isotope composition of the collected water samples was analysed by

Cavity Ring-down laser spectrometry (Picarro L2130-$i$, Picarro Inc., Santa Clara, CA, USA) in 'high precision mode' in the Laboratory of the Chair of Hydrology at the University of Freiburg, Germany. This laboratory regularly participates successfully in Water Isotope Inter-Comparisons from the IAEA (Wassenaar et al., 2018). Samples were filtered via syringe filters (0.45 µm) prior to analysis if they were muddy. Of each sample, 1 mℓ was filled into autosampler vials. According to the manufacturer's handbook, six injections per vial were analysed with the isotope analyser and raw data of the first three

injections were discarded to keep memory effects from one sample to the next at a minimum. Mean and standard deviation of the last three injections were calculated. In case there was still a memory effect and the standard deviation was larger than 0.08 ‰ for $\delta^{18}O$ or larger than 0.30 ‰ for $\delta^2H$, the fourth injection was also discarded and only the last two injections were averaged. Calibration of the raw data was then conducted using three in-house standards with distinct isotopic compositions, −14.86 ‰, −9.47 ‰, and 0.30 ‰ for $\delta^{18}O$, −107.96 ‰, −66.07 ‰, and 1.53 ‰ for $\delta^2H$ referenced to the international VSMOW-

SLAP scale (Craig, 1961a,b). The standards were analysed in triplicates each and averaged. The light and the heavy standards – embracing the samples – were used for a 2-point calibration, the third standard was used for validation. Long-term post-calibration accuracy of the validation standard was ± 0.05 ‰ for $\delta^{18}O$ and ± 0.35 ‰ for $\delta^2H$. A conservative estimate of the overall measurement uncertainty comprising sampling and analytical uncertainty yields 2.5 ‰ in $\delta^2H$ and 0.5 ‰ in $\delta^{18}O$.



## 3 Results and Discussion

The results of this study are presented in five parts, beginning with the climatological perspective on the relevance of transport pathways from the extratropics towards Barbados in Section 3.1, followed by a short overview of the atmospheric flow conditions during isoTrades in January-February 2018 in Section 3.2. The impact of the two flow regimes with extratropical influence on the stable water isotope signals, as well as on other properties of the water cycle are discussed in Section 3.3. The midlatitude dynamical flow configuration of the two extratropical transport regimes is described in a detailed comparative case

study in Section 3.4. Finally, a short summary on the link between isotope signals and the extratropical transport regimes is given in Section 3.5.

### 3.1 Importance of air with extratropical origin for the sub-cloud layer over Barbados

Even though Barbados lies close to the equator at 13.16°N, our systematic climatological trajectory analysis reveals that during the winter months January-February, the sub-cloud layer air at the BCO very rarely originates purely from the tropics. Only

for 12% of the hourly time steps, the backward trajectories never visited the subtropics or extratropics, i.e., never reached poleward of 23.5°N, in the last 10 days (Fig. 3). Fifty five percent of the air parcels arriving in the sub-cloud layer in Barbados have spent at least one day in the extratropics (north of 35°N), highlighting the climatological importance of transport pathways from the extratropics towards Barbados. Of the remaining 45%, two-thirds of the air parcels has spent at least one day in the subtropics. As shown in Fig. 3, the year 2018 is associated with an unusually large influence of air parcels originating from

relatively far north compared to the climatology. When classifying hourly time steps into the three flow regimes (see Section 2.1.2), the extratropical trade wind and dry intrusion regimes are climatologically about equally frequent (28% and 27%, Table 1), albeit with relatively large interannual variability (e.g., 62% and 32% in 2018, Table 1). Given the importance of the intertropical convergence zone (ITCZ) for the tropical circulation as well as the midlatitude jet on the subsidence from the extratropics, it is likely that their respective latitudinal positions both play an important role in shaping the interannual

variability in the residence times of air parcels in the tropics vs. the extratropics.

The timing and strength of the subsidence clearly differ in the two extratropical flow regimes (Fig. 4, see also Table 2). Extratropical dry intrusions experienced pronounced subsidence in the subtropics between 3 and 5 days before arrival, whereas in the extratropical trade wind regime the strongest subsidence occurs between 5 and 7 days before arrival (Table 2). Subsidence within the tropics, i.e., for the tropical flow regime, can also be relatively fast in rare cases, however the median

subsidence within 2 days is up to 60 hPa larger for extratropical dry intrusions four days prior to arrival (Fig. 4b). In the period 8-9 days before arrival, the subsidence rate of extratropical trade wind air parcels is similar to the one of the tropical air parcels. This might be due to the relatively strict definition of the extratropical flow regimes with some subtropical air parcels that experience a similar descent pathway as extratropical trade wind air parcels being classified into the tropical flow regime. Since the tropical flow regime is not the focus of this paper, we leave the more detailed analysis of the processes involved in

tropical flow regimes to future dedicated research.



The reasons for the difference in subsidence behaviour of the two extratropical flow regimes are related to the flow configuration over the North Atlantic (Fig. 5). The air parcels arriving in the sub-cloud layer in Barbados are advected predominantly with the typically observed easterly trade winds (see e.g. Fig. 5c, grey wind barbs for the situation in 2018). However, in some cases, a pronounced meridional transport of air occurs into the tropical lower troposphere, which interrupts

the trades in the western North Atlantic due to an extratropical perturbation as will be shown in detail in a case study in Section 3.4. Given the pronounced meridional temperature gradient in the lower troposphere in winter, such a meridional transport event is expected to occur in a slantwise descending manner towards the tropics along the sloping isentropes (or even slightly steeper due to radiative cooling). Four days prior to arrival, extratropical dry intrusion air parcels are located further north with a median cardinal direction of 66° compared to the extratropical trade wind and the tropical flow regimes, in which the air

parcels are located further east of Barbados (with a median cardinal direction of 74° and 76°, respectively). Compared to the two other flow regimes, extratropical dry intrusions thus generally follow a meridionally slantwise descending pathway towards Barbados with anomalously large subsidence rates extending over a time period going slightly beyond the four-day time window used for their definition (Fig. 4). In contrast, the extratropical trade wind regime shows anomalously weak subsidence with a substantial share of air parcels that are slightly ascending 1-2 days prior to arrival (negative subsidence in

Fig. 4b).

## 3.2 Overview of atmospheric flow conditions during isoTrades

The two transport pathways from the extratropics studied here, leave a distinct fingerprint in the isoTrades 2018 campaign mean North Atlantic flow conditions (Fig. 5). In particular, two regions of enhanced subsidence can be observed in the upper-level (320 K) and mid-level (500 hPa) vertical wind ω (Fig. 5a,b; orange contour lines): one centred over the Canary basin of

the eastern North Atlantic and the second in the western North Atlantic centred at 25°N and 55°W. These regions of enhanced upper to mid-level subsidence reflect the two transport pathways of extratropical air towards Barbados: the time-averaged peak in the vertical pressure velocity near 20°W is associated with the extratropical trade wind regime, and the peak near 55°W with the extratropical dry intrusion regime, respectively. A detailed overview of the day-to-day variability in transport between 25 January and 17 February 2018 is provided in the Supplementary material S1. Two types of large-scale features shown in

Fig. 5 play an important role in steering the air parcels from the extratropics: the midlatitude jet stream and the North Atlantic surface anticyclones.

The high surface pressure in the subtropical North Atlantic found in the campaign mean (Fig. 5c) is a well-known climatological feature arising from the presence of North Atlantic subtropical surface anticyclones. The maximum surface pressure in the eastern North Atlantic is the signature of the quasi-stationary Azores' high pressure system. The westward

extension of the zonally elongated high pressure feature in the campaign mean emerges due to more transient anticyclones, which also occur in the western part of the basin and propagate eastward from the North American east coast (see Supplementary material S1, Davis et al., 1997). The position of the surface anticyclone is strongly influenced by the position





of the midlatitude jet stream and the zonal location of anticyclonic Rossby wave breaking (ARWB) events (Thorncroft et al., 1993). Synoptic-scale ARWB events occur in situations with a meridionally extended undulation of the jet stream and appear

in the shape of meridionally elongated and narrow tongues of potential vorticity (PV) on isentropic surfaces (McIntyre and Palmer, 1984; Appenzeller and Davies, 1992). The frequency of ARWB occurrence is higher over the eastern North Atlantic (20-30%) than over the central North Atlantic (5-10%, Wernli and Sprenger, 2007; Fröhlich and Knippertz, 2008; Martius and Rivière, 2016). During ARWB, the formation of elongated PV filaments (streamers) or isolated regions with stratospheric air (i.e. stratospheric cutoffs) can dynamically induce quasi-geostrophic vertical motion in a region with ambient baroclinicity.

Quasi-geostrophic descent is typically induced at the westward side of the PV streamer or cutoff low. In the campaign mean for isoTrades in 2018, two subtropical maxima in PV on 320 K can be observed, one with PV values of 1–1.5 pvu near 50°W, 20°N and one with PV values of 1.5–2.5 pvu around 15°W, 30°N, respectively, in both cases east of the maximum in ω (Fig. 5a).

### 3.3 Contrasting conditions associated with two extratropical transport pathways towards Barbados

### 3.3.1 Contrasts in air parcel transport characteristics

In January-February 2018, the extratropical trade wind and extratropical dry intrusion regimes are both associated with ARWB over the North Atlantic, however, as discussed in Section 3.2, at different longitudes. The time of the ARWB events is estimated by searching for the period with the strongest two-day subsidence of the trajectories. For trajectories in the trade wind regime, the strongest descent of 403 hPa (2 d)$^{-1}$ occurs 6-8 days prior to arrival from 47°N, 35°W towards North Africa

(Table 2). They reach the boundary layer (i.e. the trade wind inversion) in front of the North African coast (35°N, 20°W), where they start their pathway across the North Atlantic within 6 days in the subtropical and tropical boundary layer. For trajectories in the extratropical dry intrusion regime, ARWB in the central North Atlantic leads to a maximum subsidence of on average 440 hPa (2 d)$^{-1}$ starting at 41°N, 51°W towards 29°N, 43°W (Table 2). Both regimes can thus involve rapid descent as defined by Raveh-Rubin (2017) in her dry intrusion climatology, for some air parcels exceeding 400 hPa (2 d)$^{-1}$. The

occurrence of such a rapid descent is rarer for extratropical air parcels forming a trade wind flow towards Barbados (53%) compared to air parcels forming extratropical dry intrusions as defined here (72%). This is likely due to the weaker baroclinicity in the eastern North Atlantic compared to the western part of the basin. Note that the strength of the strongest descent of 400 to 450 hPa (2 d)$^{-1}$ in the two extratropical regimes associated with ARWB is about one order of magnitude larger than the climatological vertical pressure velocity (25 hPa d$^{-1}$) resulting from adiabatic compression due to subsidence balancing

radiative cooling (~1 K d$^{-1}$, see, e.g. Holton and Hakim 2013)

The moisture source regions associated with these two regimes are shown in Fig. 6 with two maxima of moisture uptake at the southwestern edge of the anticyclone between 30°W and 60°W for the trade wind regime and between 50°W and 60°W for the dry intrusion regime. These two maxima correspond to the two maxima in time-mean surface evaporation south of the two maxima in ω (Fig. 5c). They reflect the impact of enhanced meridional subsidence of dry air towards the ocean surface, which





increases the near-surface vertical humidity gradient. The presence of an upper-level PV streamer northeast of the moisture uptake maximum in the extratropical dry intrusion regime composite (Fig. 6b) is a first indication for the role of extratropical dynamical forcing for the enhanced vertical winds, descent and moisture uptake to the northeast of the Caribbean.

The two extratropical transport regimes thus have a contrasting impact on the properties of the North Atlantic trade wind water cycle (Fig. 6, Table 2). In the trade wind regime, the sub-cloud layer air parcels are continuously warmed, due to the zonal

SST gradient across the North Atlantic, leading to continuous moisture uptake along their low-level westward pathway. These air parcels generally take up 73% of their final humidity (Table 2) from ocean evaporation and below-cloud evaporation of rainfall. The latter humidity is taken up at the south-eastern edge of North Atlantic anticyclones on average 2028 km upstream of Barbados and within 6 days prior to arrival (Table 2). Extratropical dry intrusion air parcels take up their moisture on average 1348 km upstream to the north of Barbados (Table 2, Fig. 6). After their fast adiabatic descent, the extratropical dry intrusion

air parcels reach a relative humidity of 55% and increase their specific humidity from 1.4 g kg$^{-1}$ to 4.9 g kg$^{-1}$ (Table 2). Twenty-seven percent of their humidity uptake occurs in the rapid descent phase of the air parcels and can be due to horizontal mixing or due to convective injection of moisture into the air parcels (Table 2). The rest (63%) of the humidity uptakes occurs after the period of maximum subsidence and is likely due to enhanced surface fluxes, as well as convective and turbulent mixing in the boundary layer during the slow final descent of the air parcels towards Barbados (Table 2). A more detailed study of the

physical processes involved in the moistening of extratropical trade wind compared to dry intrusion air streams is out of scope here but is planned using process-specific moisture tendency output from a simulation with the IFS model (see, e.g., Spreitzer et al., 2019).

Over the isoTrades campaign period, substantial temporal variability appears in the Lagrangian transport diagnostics shown in Fig. 7, particularly in the four-day subsidence rate and the longitudinal location of the weighted mean moisture source

location. At the beginning of the extended period of the extratropical dry intrusion in early February the residence time of the air parcels in the extratropics is particularly large with 7 days (Fig. 7). These air parcels experience a strong descent of 300 hPa (4 d)$^{-1}$ and take up their humidity relatively close to Barbados in the northeast of the island (Fig. 7). Compared to the persistent nature of the extratropical trade wind regime (prolonged periods with blue bars in Fig. 7), extratropical dry intrusions are generally more transient (short spells of red bars and short peaks in $\Delta p_{4d}$ in Fig. 7). The prolonged period of the extratropical

intrusion at the beginning of February is an exception. During the rest of the isoTrades campaign, enhanced subsidence into the sub-cloud layer associated with extratropical dry intrusions mostly occurs during short episodes. These events are often induced by a short-lived central North Atlantic upper-level PV streamer or weak cutoffs reaching particularly far south (see Supplementary material S1). Furthermore, often these events are associated with wider distributions of the Lagrangian diagnostics (shaded areas in Fig. 7), particularly for $\Delta p_{4d}$ revealing the importance of mixing of air parcels with different

transport histories.





### 3.3.2 Contrasts in local stable water isotope and meteorological conditions

The two extratropical flow regimes characterised above in terms of their transport characteristics are associated with remarkably contrasting stable water isotope signatures as will be discussed in this section along with the differences in local meteorological conditions in Barbados for the two regimes in 2018. The stable water isotope signals and meteorological

conditions during isoTrades (Fig. 8) show three striking periods: 1) the positive anomalies in the δ-values and a local minimum in $d$ between 28 and 30 January (trade wind regime case study), 2) a prolonged period of anomalously low specific humidity (12 g kg$^{-1}$), low δ-values (–76 ‰ for δ$^2$H and –12.3 ‰ in δ$^{18}$O), and high $d$ (20 ‰) between 1 and 4 February (dry intrusion case study), and 3) a second minimum in δ-values and a slightly less pronounced maximum in $d$ with specific humidity close to the campaign mean on 13 February. In Section 3.4, two comparative case studies of features 1 and 2 will be presented

focussing on the dynamical environment in which the air parcels subside towards Barbados. The third feature is a hybrid event with about half of the air parcels arriving in the sub-cloud layer showing characteristics of extratropical dry intrusions and the other half following the pathway that is typical for extratropical trade wind air parcels. This more complex event is likely also more sensitive to uncertainties in the trajectory calculation and will therefore not be further analysed in this study.

The extratropical trade wind regime leaves a distinct signature in the short-term variability of water vapour isotope signals

with anomalies of up to +4 ‰ in δ$^2$H, +1 ‰ in δ$^{18}$O, and –2 ‰ in $d$ (Fig. 8a). This regime is associated with positive anomalies in the total column water vapour, intense cold pool activity with more than 3 cold pool passages per day, leading to intense short rain showers (Fig. 8c, Table 3). The relative humidity on 28 to 30 January is high (75-85%) and strong easterly winds (8-10 m s$^{-1}$, Fig. 8b) prevail. Due to below-cloud interaction of rainfall droplets with ambient vapour, the cold pool passages leave a characteristic signature in the short-term variability of water vapour isotope signals. Campaign mean precipitation

isotope compositions are δ$^2$H$_p$=8.1 ‰, δ$^{18}$O$_p$=–0.4 ‰ and $d_p$=11.4 ‰ (Table 4), which is in the same range as the winter precipitation measured in eastern Puerto Rico (Scholl and Murphy, 2014). Rapid total re-evaporation of rain droplets (i.e. no net fractionation) e.g. at cold pool gust fronts may thus have contributed to the positive anomalies in δ values and negative anomalies in $d$ observed during the trade wind regime periods of the campaign. Furthermore, moisture input from sea spray evaporation (Thurnherr et al., 2020), a process which is certainly enhanced during extratropical trade wind flow situations due

to higher wind speeds (Table 3) can also lead to an increase in δ$^{18}$O and δ$^2$H and a decrease of $d$ in sub-cloud layer vapour.

During the particularly strong extratropical intrusion event at the beginning of February, a depletion in heavy isotopes by up to 6 ‰ in δ$^2$H, 1 ‰ in δ$^{18}$O and an increase by 6 ‰ in $d$ as was measured (Fig. 8a). In this regime, the total column water vapour is reduced by 10 mm and the lower troposphere is stabilised (+1 K in lower tropospheric stability, Klein and Hartmann, 1993) compared to the trade wind regime (Table 3). During the extratropical dry intrusion period at the beginning of February,

the lower tropospheric stability was particularly large (~16 K, Fig. 8c). The strong drying of the free troposphere during such an event (Fig. 10b) is likely to enhance radiative cooling at the top of the boundary layer, which thereby strengthens the inversion (Bony et al., 2020a). There is hardly any cold pool activity and large-scale areas of clear-sky conditions dominate (–9% in total cloud cover compared to the conditions during extratropical trade winds, Table 3) due to the continuous supply





of dry upper-level extratropical air into the tropical lower free troposphere. The entrainment of the dry air into the boundary
layer lowers the surface humidity at the BCO (–2% in RH and –0.6 g kg$^{-1}$ in $q$ in the composite mean compared to the campaign
mean in Table 3) thereby increasing the near surface vertical humidity gradient, and intensifying ocean evaporation (+0.5 mm
d$^{-1}$). A clear minimum in the BCO wind speed can be observed (2-4 m s$^{-1}$, Fig. 8b) as well as a very low near-surface relative
humidity of 55% compared to the campaign mean of 71% (Fig. 8b). No precipitation was registered at the BCO for a period
of 5 days during the extratropical dry intrusion case at the beginning of February (Fig. 8b). The combination of these processes
led to an increase of near-surface $d$ due to enhanced non-equilibrium fractionation effects during ocean evaporation and a
lowering of the δ in water vapour.

Concluding this section, we would like to note that, even though many processes affect the variability of water vapour isotopes,
the clear contrast in isotope signals on our chosen case study days provides a robust foundation for their use as proxies for the
two transport pathways from the extratropics in 2018.

### 3.4 Involved dynamical processes: a comparative case study

On the extratropical trade wind regime day (29 January), the moist trade wind layer is particularly deep with nearly saturated
air and easterly winds reaching up to 600 hPa (Fig. 9a). The shallow cumulus cloud pattern on that day features gravel-like
structures with many cold pools developing in the vicinity of Barbados (Fig. 10a,b). The radio sounding from 2 February (Fig.
9b) illustrates again the much drier near surface conditions, the stronger northerly wind component up to 400 hPa and enhanced
lower tropospheric stability (between 950 hPa and 600 hPa) typical for extratropical dry intrusions. In this regime, cloud free
conditions are prevailing in a large area around Barbados (Fig. 10c,d). The dynamical environment in which the air parcels
arriving on these two exemplary days travel towards Barbados is analysed in more detail in this section.
Both on 29 January and 2 February, the air parcels arriving in the upper troposphere above Barbados (trajectories with red and
orange dots in Fig. 11) are associated with an approximately zonal westerly flow. However, for air parcels arriving at low
levels, the transport pathways on the two days strongly differ. On 29 January, representative of the extratropical trade wind
regime, sub-cloud and cloud layer air parcels (blue and green) descend in front of the western European and North African
west coast in the context of an eastern North Atlantic surface anticyclone (Fig. 11a, Fig. 12a[1]). This anticyclone is associated
with an ARWB that develops over western Europe and the Mediterranean. The air parcels that descend fastest are located in
an area with enhanced subsidence at the southwestern tip of a PV streamer that moves over the Atlas (Fig. 12a and the
animation in the Supplement S2). After their descent during the wave breaking along the northwest African coast and over
North Africa, the air parcels travel across the North Atlantic within the boundary layer towards Barbados.

---

[1] For the sake of readability only air parcels with arrival times every 3 hours starting at 00 UTC are shown.



The extratropical dry intrusion air parcels arriving in Barbados between 1 to 4 February subside in the context of a central North Atlantic ARWB (Fig. 13[1], and the animation in Supplement S3). These air parcels descend slantwise from the north

within an airstream that consists of two branches (Figs. 11b and 13):

-    Branch 1 (B1) includes a few air parcels that come from the north and the east at low levels and ascend rapidly (Fig. 13a-d), likely due to deep convection, in the vicinity of the PV streamer that then develops into a PV cutoff on 28 January near 47°W, 30°N in the vicinity of the line C1 in Fig. 11b. Subsequently, these air parcels descend towards Barbados at the western edge of the PV cutoff (see animation in Supplement S3).

-    Branch 2 (B2) with a majority of air parcels that first travel within the midlatitude jet (Fig. 13a) and then subside meridionally in an area with enhanced subsidence at the right jet exit (Fig. 13a,b and animation in Supplement S3), and a few days later west of the PV cutoff (in the vicinity of line C2 in Fig. 11b, Fig. 13c,d).

The air parcels from B1 thus experience strong ascent at the eastern edge of the PV streamer (blue dots in Fig. 13c), whereas the air parcels from B2 experience strong descent at the western flank of the PV streamer (red dots in Fig. 13c) and subsequently

of the PV cutoff (red dots in Fig. 13d). Interestingly, the two branches join along the PV streamer and the air parcels from B1 and B2 gather below and around the PV cutoff between 28 and 29 January (Figs. 13d, 14a,c). Two cross sections along lines C1 and C2 in Fig. 14 illustrate the environment in which the air parcels from B1 and B2 gather and achieve their concerted final meridional descent into the tropics.

The dynamics of the large-scale descent of the air parcels arriving in Barbados in the two case studies is influenced by several

extratropical weather systems developing in the context of the ARWB events. For example, during the extratropical dry intrusion, a large area of enhanced subsidence can be observed near the eastern edge of the transient western North Atlantic anticyclone (Fig. 11b) and west of an extratropical cyclone located below the PV cutoff on 29 January (Fig. 11b).

The different zonal location of the two ARWB events studied here has a distinct impact on the evolution of the thermodynamic properties of the air parcels associated with the extratropical trade wind and dry intrusion cases. Figure 15 shows the

Lagrangian evolution of the two contrasting cases in the phase space of temperature ($T$) vs. potential temperature ($\theta$). Visualizing the evolution of trajectories in this $\theta$ - $T$ phase space serves to distinguish adiabatic vs. diabatic processes that occur along the flow (e.g., Bieli et al., 2015; Papritz et al., 2019). The differences in the thermodynamic behaviour of the sub-cloud layer air parcels from the extratropical trade wind and dry intrusion flow regime case studies can thereby be summarised. The descent voyage of the extratropical dry intrusion shown by the red line in Fig. 15 can be split into three stages. During the

first stage, 5-10 days before arrival, the air parcels are moving relatively fast within the midlatitude jet and the thermodynamic properties of the air parcels remain approximately constant. During the second stage, 3-5 days before arrival, rapid descent (200 hPa (2 d)$^{-1}$, Fig. 15, red line) occurs and the airstream warms adiabatically in the upper troposphere (motion towards the right in Fig. 15 at relatively constant $\theta$ of about 304 K) in a largely cloud-free region (Fig. 14a,c and Fig. 15 red line). During the third stage of the descent, starting three days before arrival, the sinking air is further warmed adiabatically due to the

descent, but also cooled diabatically in particular during nights (motion towards the bottom right in Fig. 15). In this third stage, the air parcels are located at the top of tropical low-level clouds (Fig. 14b,d), where they are most probably diabatically cooled





by either microphysical (cloud evaporation) or radiative (inversion or cloud top radiative cooling) processes. In this stage, the air parcels start to take up substantial amounts of water vapour (Fig. 14d, Fig. 15 dots coloured with specific humidity). In contrast, the extratropical trade wind airstream (blue line in Fig. 15) begins to be moistened seven days before arrival and is

continuously diabatically heated by surface fluxes within the sub-cloud layer as it travels across the North Atlantic during the five days before arrival at the BCO.

In summary, for both case studies, ARWB plays a key role in setting the scene for a rapid descent of air parcels that either reach Barbados directly from the north in the case of central North Atlantic ARWB (dry intrusion), or from the east in the case of eastern North Atlantic ARWB (trade wind) after a six day voyage within the subtropical and tropical boundary layer.

This contrast in the location of ARWB and in the pathway of the air parcels into the trades strongly influences the thermodynamic evolution of the air parcels arriving at the BCO near the surface.

### 3.5 Linking the deuterium excess to the moisture transport pathways and cloud patterns around Barbados

During isoTrades, the above discussed variability in moisture transport pathways and local conditions lead to two very interesting summarising relations: 1) between the $d$ and the moisture source distance (Fig.16a), and 2) between $d$ and the cloud

patterns (Fig. 16b). These relations and their significance are shortly discussed in the following.

A strong anticorrelation between the $d$ and the distance to the moisture uptake region is found during isoTrades (Fig. 16a). The smaller distance to the source during extratropical dry intrusions (500-1000 km) is indicative of enhanced and rapid moisture uptake (because of the large humidity deficit in the rapidly subsiding air) with little time for interaction with rainfall and clouds underway. Much longer transport distances are associated with the trade wind flow regime, in which longer-range transport

across an increasing SST gradient is favourable for the formation of clouds and rainfall underway, leading to below-cloud interaction between sub-cloud layer vapour and falling rain drops. The $d$ therefore can be seen as a measure for the time (and place) of the air parcels' entry into the boundary layer with high $d$ indicating "old" boundary layer air and high $d$ "new" boundary layer air of extratropical origin. A clear observational linkage between the extratropical transport pathways and the isotope signals from the BCO is thereby obtained.

From the above detailed analysis of the impact of the different transport regimes on the trade wind water cycle, the question arises whether different transport pathways favour the occurrence of specific cloud patterns. One way to discriminate among different cloud patterns around Barbados using the surface wind speed and the lower tropospheric stability has been presented in Bony et al., 2020b. The dominant cloud patterns during the isoTrades campaign were gravel and fish with some occurrence of sugar (Fig. 16b, Supplement S1, see also Stevens et al., 2020a). The high $d$ anomalies during the extratropical dry intrusions

are associated with the fish cloud pattern, while the low $d$ anomalies are associated with the gravel cloud pattern. This provides a promising starting point for a more detailed investigation on the importance of the extratropical origin of air parcels during prolonged episodes with fish clouds, which will be performed using the EUREC[4]A isotope datasets.





## 4 Conclusions and Outlook

In this paper, we show that in winter, the dynamics of the large-scale descent in the subtropics is essential for the variability
of the stable water isotopes, the sub-cloud layer humidity, and the low-level cumulus cloud cover in the western North Atlantic
trade wind region. The stable water isotope signals in vapour and precipitation from a 24-day measurement campaign in
January and February 2018 at the Barbados Cloud Observatory (BCO) are used as tracers to study the properties of two
transport regimes associated with extratropical dynamics: the extratropical trade wind flow (61% occurrence frequency in
2018) and extratropical dry intrusions (32%). In addition, in 2018, only 6% of the times were classified into the (sub)tropical
flow regime. The two extratropical flow regimes are associated with distinct transport pathways and thermodynamic
evolutions, which we identified by using backward trajectories calculated with three-dimensional ERA5 wind fields for 2018
and in a climatological analysis for 2009–2018. Given the generally small variability of the water isotope composition at the
BCO ($\delta^{18}$O = (–10.7 ± 0.4) ‰, $\delta^2$H = (–71.9 ± 2.0) ‰, $d$=(14.3 ± 2.1) ‰), high precision measurements with a fast-response
cavity ring-down laser spectrometer were necessary to resolve the differences in the isotope signature of the two extratropical
flow regimes.

A climatological analysis of the air parcel origin in January-February 2009–2018 revealed that 1) air parcels arriving in the
eastern Caribbean above Barbados mainly descend from the extratropics (55%) with two alternating subsidence pathways in
the central and eastern North Atlantic, respectively, and 2) the occurrence frequency of the extratropical dry intrusion regime
is climatologically similar (27%) to the occurrence frequency of the extratropical trade wind regime (28%). There is large
interannual variability in the contribution of air parcels from the extratropics to sub-cloud layer air above Barbados, as well as
in the occurrence frequency of the different flow regimes. The reason for this large variability is likely related to the dynamics
of Rossby wave breaking along the midlatitude jet stream and the latitudinal location of the ITCZ.

We show that in January-February 2018, the extratropical trade wind and dry intrusion regimes are both associated with
anticyclonic Rossby wave breaking (ARWB) over the North Atlantic. The longitude of the ARWB determines which of the
two pathways is active. ARWB in the eastern North Atlantic close to the West African coast leads to the formation of a low-
level easterly trade wind flow towards Barbados. The air parcels in this first regime typically start their descent eight days
before arrival at 47°N, 35°W, and reach the boundary layer in front of the North African Coast (35°N, 20°W) in the context
of an eastern North Atlantic anticyclone six days before arrival. These air parcels then cross the North Atlantic at low levels
and their specific humidity increases substantially from about 6 to 10 g kg$^{-1}$ during this passage. A deep layer establishes with
easterly winds from the surface up to ~600 hPa above Barbados such as on 29 January 2018.

In contrast, ARWB over the central North Atlantic favours the occurrence of descending extratropical dry intrusions directly
into the sub-cloud layer of Barbados. After exiting the upper-level extratropical jet stream in the right exit region near 41°N,
51°W, these extratropical air parcels start their three-stage equatorward descent six days prior to their arrival in Barbados. The
first rapid descent stage is mainly adiabatic during which the largest part of the descent of 440 hPa (2d)$^{-1}$ towards 29°N, 43°W
is accomplished. During the slower second stage, the air parcels are located on top of tropical low-level clouds, they also





experience diabatic cooling compensating some of the adiabatic warming, as shown for the case of 2 February 2018, and they reach the sub-cloud layer in Barbados within four days. During this period, their specific humidity increases very strongly from about 3 to 10 g kg$^{-1}$. For extratropical dry intrusion air parcels to directly reach Barbados from the North, key ingredients are a quasi-stationary, central North Atlantic upper-level PV cutoff formed during the ARWB, potentially coupled to a

subtropical surface cyclone underneath. The combined influence of the surface cyclone and the upper-level cutoff is to steer the subsiding air of the dry intrusion towards the Caribbean. The two regions of preferred ARWB and descent from the extratropics leave a distinct imprint also in the campaign time-mean field of mid-tropospheric vertical motion over the North Atlantic.

The two extratropical transport regimes have a contrasting impact on the low-level cloud patterns, sub-cloud layer humidity

and stable water isotope properties. In the extratropical trade wind regime, the sub-cloud layer air parcels take up 73% of their final humidity from ocean evaporation and below-cloud evaporation of rainfall near the south-eastern edge of North Atlantic anticyclones on average 2028 km upstream of Barbados and within six days before arrival. Due to below-cloud interaction of rain droplets with ambient vapour, cold pool passages leave a distinct signature in the short-term variability of water vapour isotope signals with anomalies of up to +4 ‰ in $\delta^2$H, +1 ‰ in $\delta^{18}$O, and –2 ‰ in $d$, compared to the campaign mean. The

important role of cold pools in the extratropical trade wind regime is reflected in their gravel-like signature on visible satellite images. In the extratropical dry intrusion regime, the air parcels take up their moisture on average 1348 km upstream of Barbados. A smaller share of this moisture uptake (27%) occurs during the rapid descent stage due to convective and turbulent mixing of moisture into the descending airstream and the main uptake (63%) occurs during the slow descent stage when the dry air approaches and interacts with the cloud-topped marine boundary layer. The total column water vapour is smaller by 6

mm compared to the trade wind conditions, leading to large-scale areas of clear sky conditions (7% reduction of total cloud cover compared to campaign mean conditions) due to the continuous supply of dry upper-level air into the tropical lower free troposphere. The penetration of the dry air parcels into the boundary layer lowers the humidity (–2% in RH and –0.6 g kg$^{-1}$ in $q$ compared to the campaign mean), thereby increasing the near-surface humidity gradient, and intensifying ocean evaporation (+0.5 mm d$^{-1}$). The combination of these processes leads to a decrease in heavy isotopes by 6 ‰ in $\delta^2$H, 1 ‰ in $\delta^{18}$O and an

increase by 6 ‰ in $d$, as was measured for a particularly strong extratropical dry intrusion event described in a detailed case study.

The results of this study on the dynamics of atmospheric moisture uptake in the extratropical trade wind regime raise the question, whether the extratropical origin of these air parcels is relevant for their subsequent evolution. The thermodynamic properties of these air parcels are mainly determined within the trades, the dominant part of the moisture uptake occurs in the

subtropical and tropical North Atlantic within 6 days prior to arrival and only a very small part of the remaining moisture in Barbados actually originates from the extratropical upper troposphere. If the properties of the trade wind air masses are determined only after the entry of the air parcels into the trade wind region, their exact descent pathway prior to reaching the trades might be irrelevant. To more thoroughly address this question, a more detailed analysis of the cloud patterns associated with the trade wind regime is needed. It has yet to be investigated if air parcels with a descent history associated with an eastern





North Atlantic ARWB event lead to different cloud patterns compared to tropical air parcels that descend over Africa as outflows from the Intertropical Convergence Zone (ITCZ) deep convective systems. Deep convective outflow air parcels from the ITCZ might be moister and their subsidence rate, which is mainly controlled by radiative cooling (Salathé and Hartmann, 2000), might be smaller compared to the rapidly subsiding eastern ARWB air parcels. Such tropical origin air parcels would therefore probably take up less humidity within the trades. These reflections lead us to the overarching question about the size

of the spatio-temporal window within which the atmospheric circulation is relevant for the observed variability in the cloud cover and lower tropospheric humidity in the North Atlantic trades. Although the typical moisture residence time in the atmosphere is around 4-5 days in this region (Sodemann, 2020), the lifetime of weather systems within which the airstreams take up their humidity tends to be longer, in particular in the case of eastern subtropical anticyclones. Future, more detailed investigations will be needed to shed light on the relevance of the tropical vs. extratropical air parcel origin fuelling the winter

North Atlantic trade wind layer.

  Episodes with Saharan dust transport across the North Atlantic reaching the Caribbean are rarer in winter than in summer (Gläser et al., 2015) but might nevertheless play a role for the cloudiness over Barbados in January and February (Gutleben et al., 2019). Such events can be associated with eastern North Atlantic ARWB (Knippertz and Fink, 2006). During EUREC[4]A, multiple days with Saharan dust arriving above Barbados have been observed. Aircraft-based lidar (Chazette et al., 2020,

Stevens et al., 2020b) and stable water isotope measurements have been performed in this period, which provides the data basis to investigate this aspect in more detail.

  In addition to the above mentioned open question on the relevance of the extratropical origin of the air in the trade wind regime, other caveats of this study are associated with the limited spatio-temporal coverage of the performed measurements and the subjectivity of the thresholds of subsidence and extratropical residence time involved in the flow regime definition. The results

presented in this paper are based on a relatively short campaign including only near-surface measurements. The isoTrades dataset already shows large variability in the transport patterns and local conditions at the BCO depending on the location of the ARWB over the North Atlantic. An extension with data from different surface and airborne platforms during EUREC[4]A in January-February 2020 as well as longer time series of isotopes measured at the BCO, including also other seasons, will be very valuable. Furthermore, here only the flow regime and thermodynamic history of sub-cloud layer air parcels has been

studied. However, the vertical thermodynamic structure of the lower troposphere is also strongly influenced by differential advection at higher levels. A future study on the impact of the different transport histories of air parcels arriving at various vertical levels in the troposphere is planned, using the comprehensive balloon sounding array compiled during EUREC[4]A (Stephan et al., 2020).

  Finally, it remains open to what extent low-level cloud formation modulates the isotope signature of the sub-cloud layer vapour

forming the cloud from convective updrafts and how much of the vapour isotope signature of the large-scale transport regime is still reflected in precipitation reaching the surface. Previous studies have identified some influence of large-scale advection and moisture source signals in orographic clouds over land (e.g. Spiegel et al., 2012, Scholl et al., 2014). Measurements of the short-term isotope variability in low-level clouds and shallow convective precipitation are however very scarce.



In summary, this paper provides new insight into the role of the large-scale circulation for the environment in which trade

wind cumulus clouds form and thereby contributes to the ongoing climate research effort to elucidate the role of the coupling

between clouds and the circulation. Furthermore, this study underlines the importance of extratropical dynamics for the

humidity and the isotope signature of the tropical trade wind region in winter.

*Data availability.* The ERA5 reanalyses used in this study can be accessed from the ECMWF website

(https://www.ecmwf.int/en/forecasts/datasets/reanalysis-datasets/era5). The imagery of the Earth Observing System Data and

Information System (EOSDIS) can be obtained from the Worldview Snapshots application (https://wvs.earthdata.nasa.gov).

The isoTrades dataset is published in the ETH data collection, available online here: 10.3929/ethz-b-000439434.

*Author contributions*. F.A. initiated the project in exchange with B.S. and S.B.. P.G., F.A., F.J. and R.V. were involved in the

field work on Barbados. P.G. and F.A. carried out the stable water isotope measurements during isoTrades. F.J. provided the

meteorological data, R.V. the cold pool statistics. F.A. performed the post-processing of the water vapor isotope data, the

trajectory-based analysis and wrote the paper. F.A., H.W., F.D., and L.V. discussed the climatological trajectory analysis. All

co-authors contributed to the interpretation of the results and commented on the manuscript.

*Competing interests*. The authors declare that they have no conflict of interest.

*Acknowledgements*. The isoTrades campaign received funding from the ETH Seed Grant No SEED-0517-2, F.D.

acknowledges funding from the German-Swiss project "MOisture Transport pathways and Isotopologues in water Vapour

(MOTIV)" supported by the Swiss National Science Foundation Grant No. 164721, L.V. has received funding from the Swiss

National Science Foundation Grant No. 188731. Insightful simulations that helped in the preparation phase of the isoTrades

campaign could be conducted at the Swiss National Supercomputing Centre (CSCS) within the small production project sm08

2017-2018. Marco Vecellio is thankfully acknowledged for crafting a beautiful and robust inlet. We thank Silvia Nast (ETH

Zurich) and Marvin Forde (Barbados Institute of Meteorology and Hydrology) for their precious help with logistics and

customs. Barbara Herbstritt (University of Freiburg im Breisgau) performed the measurements of the liquid samples (reference

standards and precipitation samples). F.A. thanks Shira Raveh-Rubin for inspiring discussions. The authors acknowledge

MeteoSwiss and ECMWF for the access to the ERA5 reanalyses and the use of imagery from the Worldview Snapshots

application (https://wvs.earthdata.nasa.gov), part of the Earth Observing System Data and Information System (EOSDIS).

**Appendix A: Justification of flow regime classification framework**

The flow regime classification is performed separately every hour in January-February in the period 2009 to 2018 based on

the 40 sub-cloud layer trajectories. If the median residence time north of 35°N ($\tilde{\tau}_{35}$) of the air parcels for a given time step is

$\geq 1$ day, the time step is classified into the **extratropical flow regime**. If $\tilde{\tau}_{35} < 1$ day, the hourly time step is classified into the

**tropical flow regime**. Note that air parcels having spent several days in the subtropics are also assigned to the tropical flow

regime. The extratropical flow regime is further stratified based on the median amplitude of the subsidence in the four days

prior to arrival ($\widetilde{\Delta p}_{4\mathrm{d}}$) of the air parcels at the BCO. If $\widetilde{\Delta p}_{4\mathrm{d}} \geq 100$ hPa (4 d)$^{-1}$ the date is classified as being influenced by an

**extratropical dry intrusion**. If $\widetilde{\Delta p}_{4\mathrm{d}} < 100$ hPa (4 d)$^{-1}$, the date is classified into the **extratropical trade wind flow regime**

with sub-cloud layer air originating from the extratropics but having experienced limited subsidence in the four days prior to

arrival in Barbados. The threshold value for the subsidence rate was chosen close to the ERA5 climatological value of free

tropospheric subsidence (at 500 hPa) in the region of Barbados. This corresponds approximately to the vertical pressure



velocity that is expected from subsidence balancing radiative cooling ($\sim$25 hPa (d)$^{-1}$, with $\sim$1 K(d)$^{-1}$ longwave radiative cooling, see Holton and Hakim, 2013). This choice is based on the hypothesis that air parcels penetrating into the sub-cloud layer due to large-scale advection with a subsidence rate that exceeds its climatological free tropospheric value are likely to have a considerable impact on the sub-cloud layer properties. The time window for defining the subsidence threshold was chosen

based on the typical lifetime of water vapour in the subtropics and winter tropics of four days (Sodemann, 2020). The subsidence rate within four days thus seems to be an adequate choice to investigate the impact of the different flow regimes on the western North Atlantic trade wind water cycle.

Note that in comparison to the definition of dry intrusions in Raveh-Rubin (2017), with a slantwise subsidence rate of > 400 hPa (2 d)$^{-1}$, the extratropical dry intrusions defined here subside much more slowly. For our purpose, the important point

is that our air parcels classified as extratropical dry intrusions show a pronounced subsidence within the time window that is relevant for the water vapour lifetime in this region. In addition, for comparing with the climatology of Raveh-Rubin (2017), the percentage of hourly time steps with at least one backward trajectory from the BCO with a descent of > 400 hPa (2 d)$^{-1}$ is computed for the two extratropical flow regimes.

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





(a)

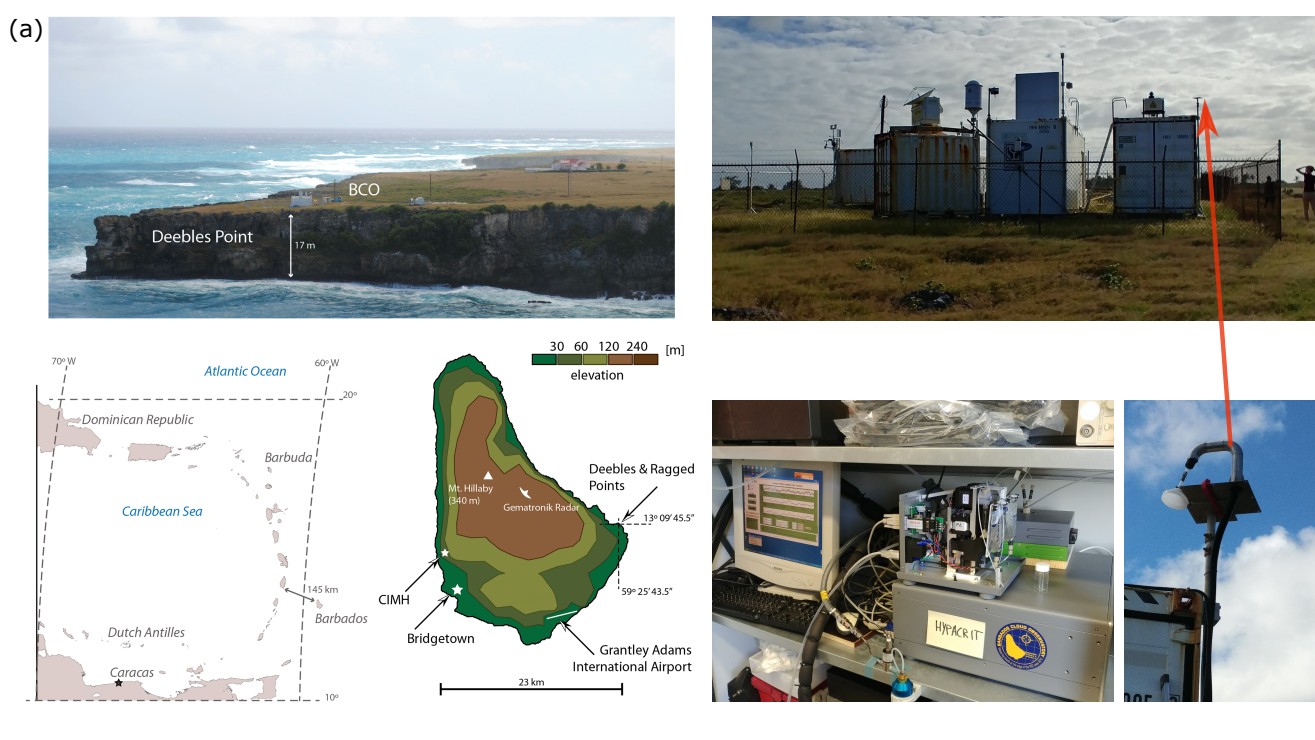

(b)

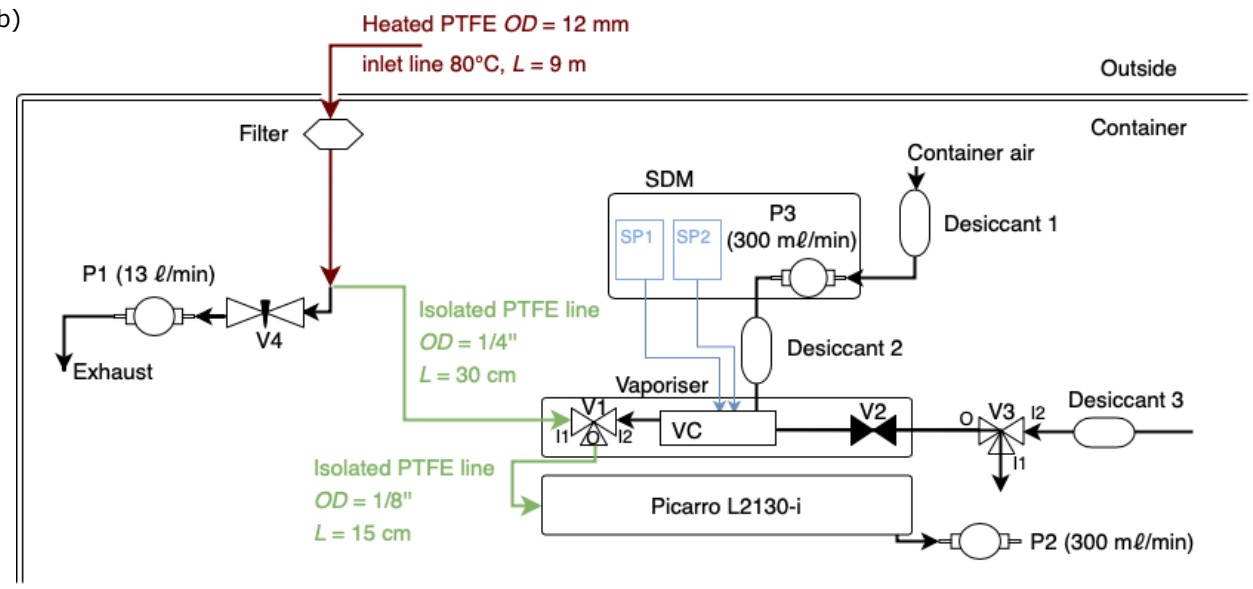

**Figure 1: Setup of the water vapour isotope measurement system at the Barbados Cloud Observatory, BCO. (a) The situation of the island of Barbados in the North Atlantic at the eastern boundary of the Caribbean Sea, together with a picture of the BCO at Deebles point, one of the easternmost landmark of the island (from Stevens et al., 2016). On the right, the measurement container with the inlet and the Picarro L2130 cavity ring-down spectrometer inside the container. (b) Gas flow schematic with P1-3 air pumps, SP1-2 high precision liquid pumps for the injection of standard liquid water into the vapouriser, V1-3 three-way valves, V4 a flow restrictor, SDM is the standard delivery module of Picarro, VC is the vapouriser chamber in which the calibration gas is prepared.**





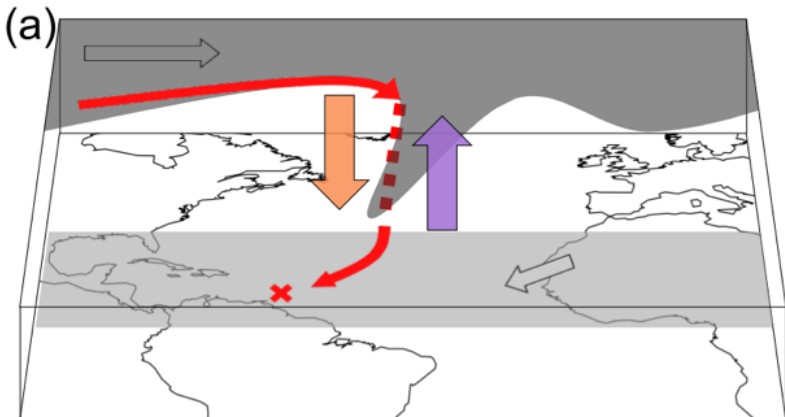

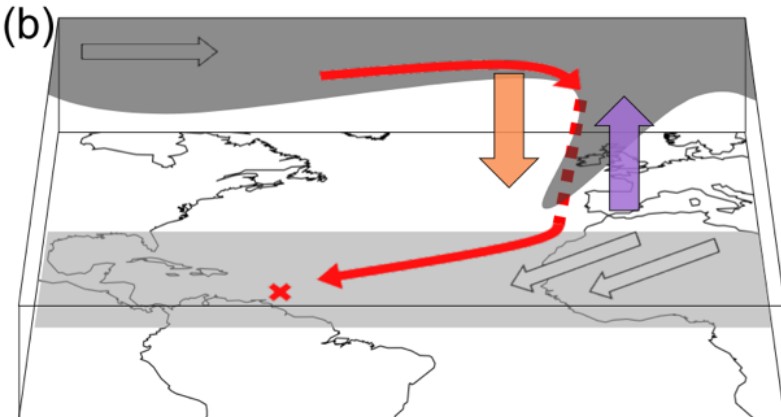

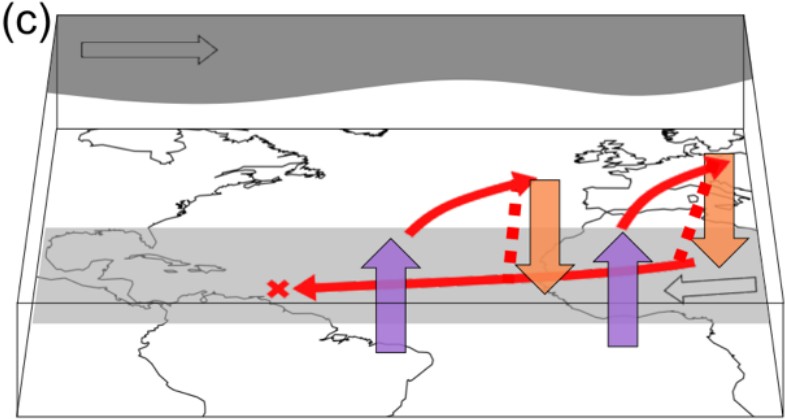

Figure 2: Three-dimensional schematic of the three flow regimes for air parcels arriving in the sub-cloud layer ($p \geq 940$ hPa) in Barbados. The extratropical dry intrusion regime is shown in (a), the extratropical trade wind regime in (b), and the tropical regime in (c). The upper-level westerly flow (e.g. on the 310 K isentrope) is indicated with the dark grey surface, the low-level trade wind flow (900-1000 hPa) with the light grey surface, strong descending motion is shown by orange arrows, strong ascending motion is shown by violet arrows, the red line indicates the air parcels' typical trajectory in each regime with dashed red representing rapid descent.


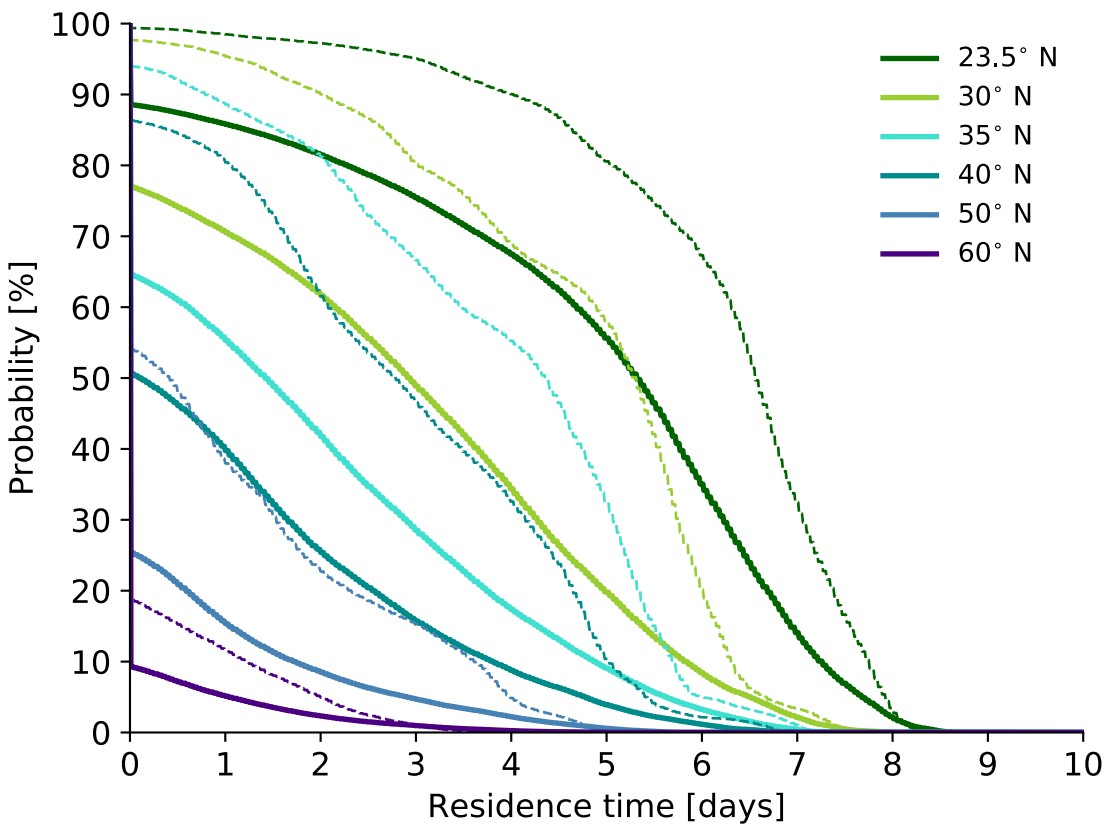

**Figure 3: Exceedance probabilities of the residence time in different spherical caps (north of 23.5°N, 30°N, 35°N 40°N, 50°N, and 60 °N) of air parcels arriving in the sub-cloud layer ($p \geq 940$ hPa) in Barbados, based on a climatology of 10-day backward trajectories from Barbados calculated with ERA5 in January and February 2009 to 2018 (568'320 trajectories, solid lines) and for January and February 2018 (isoTrades campaign; 23'040 trajectories, dashed lines).**







**Figure 4: Climatological mean and standard deviation (bar) showing the pressure (a) and two-day subsidence rate (b) of the 40 sub-cloud layer air parcels within 1 to 10 days prior to their arrival above Barbados in January-February 2009-2018, classified into regimes with different transport pathways. The extratropical dry intrusion regime (27% occurrence frequency) is shown in dark red, the extratropical trade wind regime (28%) in blue, and the tropical regime (44%) in yellow.**





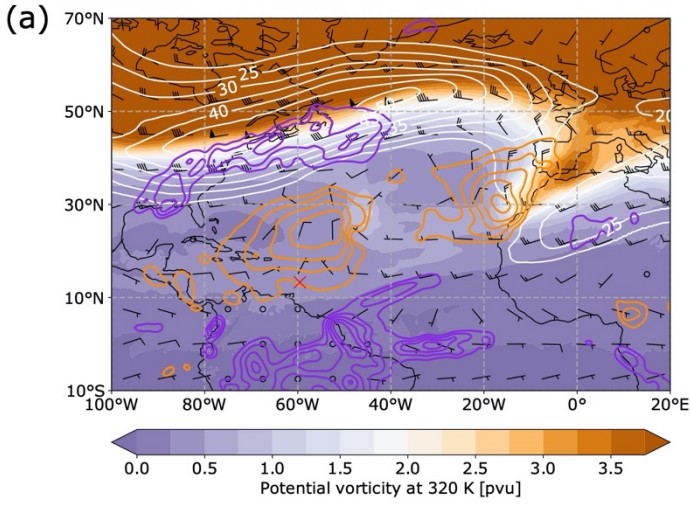

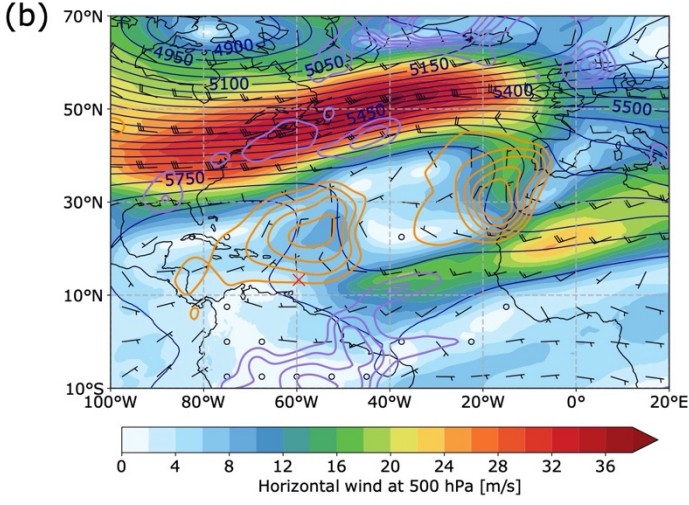

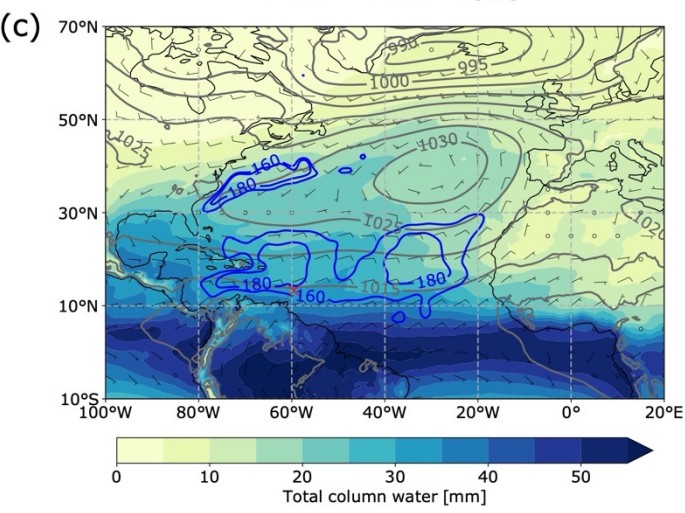

**Figure 5: Overview of atmospheric flow conditions, time-averaged for the period 25 January to 17 February 2018 from hourly ERA5 reanalyses. (a) upper-level flow conditions on 320 K with potential vorticity (colours), horizontal wind speed (white contours from 20 to 50 m s$^{-1}$ in steps of 5 m s$^{-1}$), wind barbs (black), and vertical wind ω (orange contours for descent 0.04 to 0.1 Pa s$^{-1}$ in steps of 0.02 Pa s$^{-1}$ and violet contours for ascent -0.1 to -0.04 Pa s$^{-1}$ in steps of -0.02 Pa s$^{-1}$). (b) flow conditions at 500 hPa with horizontal wind speed (colours) and geopotential height (blue contours). Wind barbs and vertical winds at 500 hPa as in (a). (c) near-surface flow conditions with sea level pressure (grey contours), wind barbs at 900 hPa (grey), accumulated surface evaporation E during the period (blue contours, in mm, positive upward), and average total column water (colours). The red cross indicates the position of Barbados. The contours of ω and E were smoothed with a Gaussian filter for better readability of the figure.**

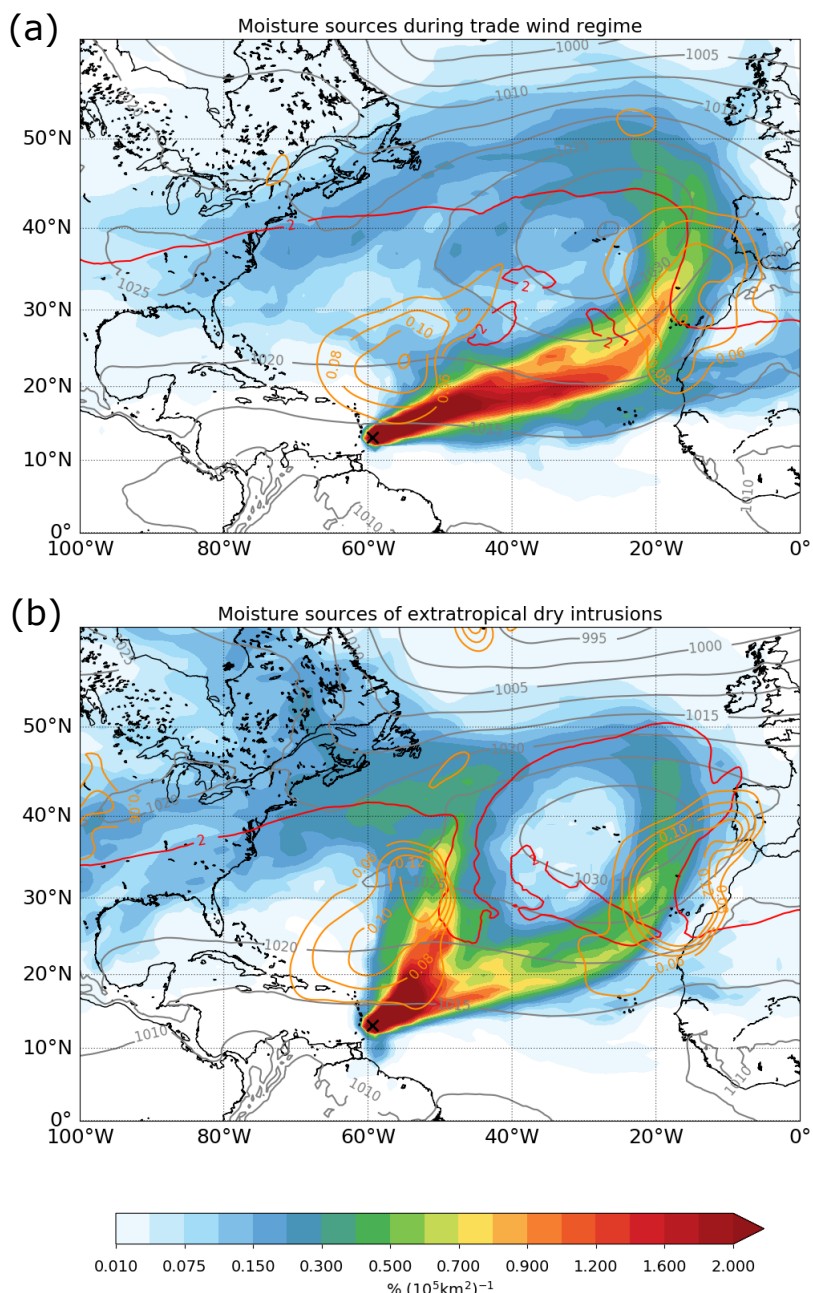

**Figure 6: Moisture sources during the time period from 25 January to 18 February, separately for (a) the extratropical trade wind regime (62% occurrence frequency in 2018) and (b) the extratropical dry intrusion regime (32% occurrence frequency in 2018). The percental contribution of uptakes per $10^5$ km$^2$ to the final specific humidity in the boundary layer at the BCO is shown in colours. The composite 2 pvu isoline on the 330 K isentrope is shown by the red contour line. Regions of subsidence averaged during the regimes are shown by orange contours of vertical wind ω at 500 hPa (in Pa s$^{-1}$). The grey contour lines show the regime averages of sea level pressure in hPa. The composite fields of sea level pressure, PV on 330 K and ω are shown for 4 days prior to the arrival of the air parcels in Barbados.**

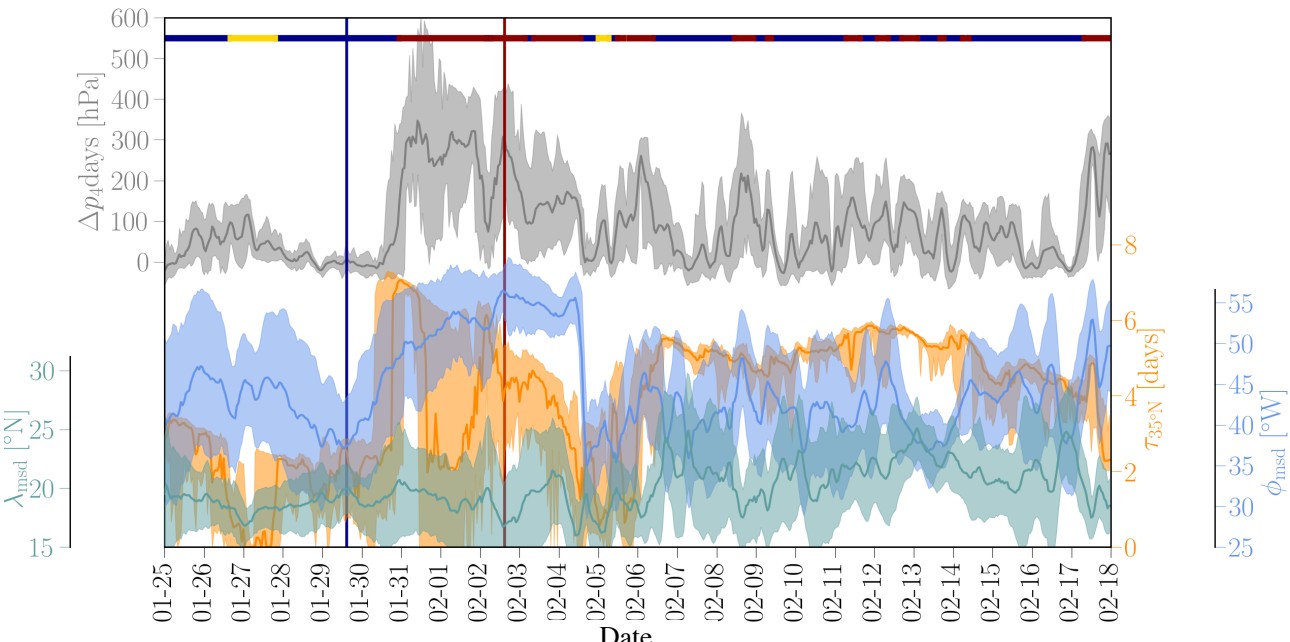

**Figure 7: Lagrangian diagnostics for air parcels arriving in the sub-cloud layer ($p \geq 940$ hPa) above Barbados. The slantwise subsidence in 4 days, $\Delta p_{4d}$ in grey corresponds to the difference between the arrival pressure and the pressure of the air parcels 4 days before arrival. The residence time of airmasses in the extratropics $\tau_{35}$ in the 10 days prior to their arrival is shown in orange. The weighted mean moisture source longitude ($\phi_{msd}$, blue) and latitude ($\lambda_{msd}$, turquoise) represent the centre of mass of the area over**

950 **which moisture was taken up by the air parcels, the shaded area shows the weighted standard deviation of the mean. The grey and orange lines for $\Delta p_{4d}$ and $\tau_{35}$ represent the medians and the shaded area the 10th and 90th percentiles of the respective distributions for the 40 trajectories per hourly time step. The coloured bars at the top indicate the regime classification with blue for the extratropical trade wind regime, dark red for the extratropical dry intrusion regime, and yellow for the tropical regime. The vertical lines indicate the time steps chosen for the case studies in Section 3.4 at 15 UTC 29 January 2020 and at 15 UTC 2 February 2020,**

**respectively. The cloud patterns for these time steps are shown in Fig. 10 and the trajectories in Fig. 11.**



**Figure 8: Hourly stable water vapour isotope measurements in ambient air (a), meteorological variables from the BCO, and ERA-5 data averaged over a box with horizontal extent 13°-15° N and 58-60° W (c) from 25 January to 17 February 2018. *T* is the air temperature, *RH* the relative humidity, *U* the wind speed, *D* the wind direction, *P* the rainfall rate, *TCW* for total column water, *E* for surface evaporation (positive upward) and *LTS* for lower tropospheric stability (*LTS*=$\theta_{700\,hPa}$-$\theta_{1000\,hPa}$, with $\theta$ potential temperature in K). The coloured bars at the top indicate the regime classification as in Fig. 7.**

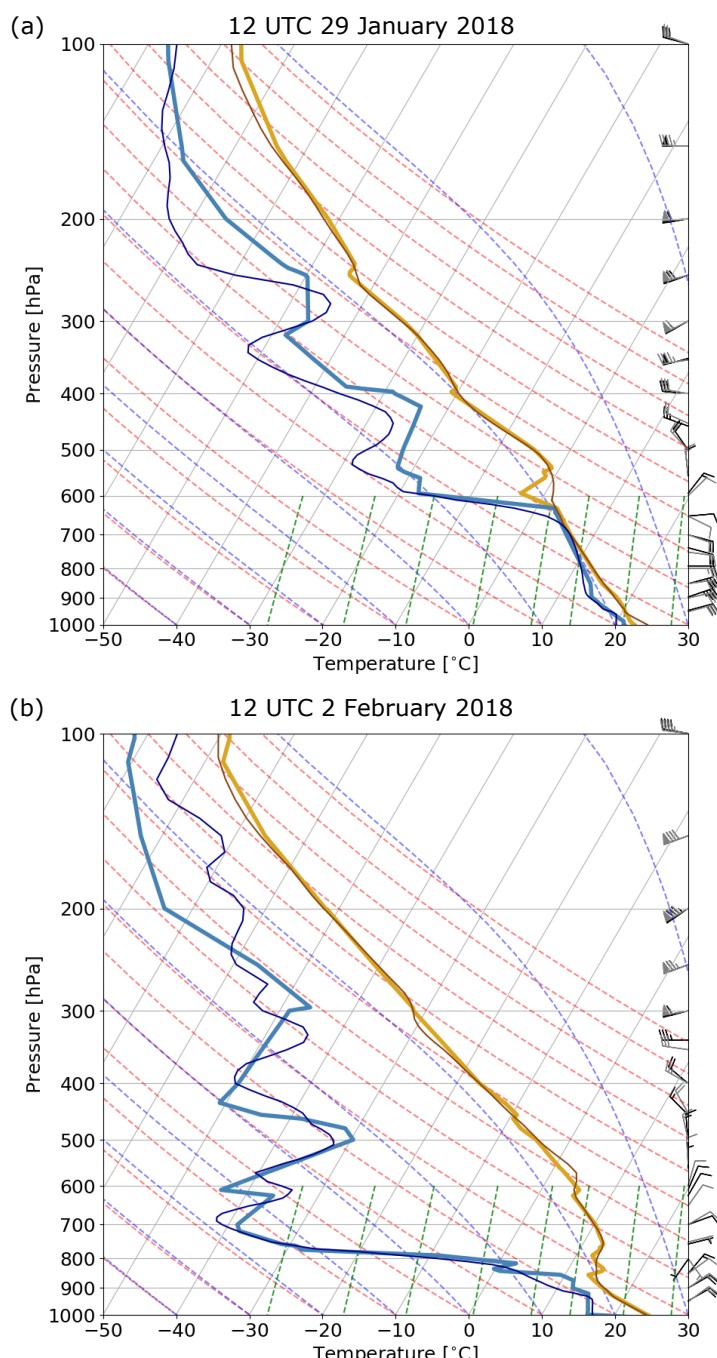

**Figure 9: Vertical soundings from Barbados Grantley Adams Airport at (a) 12 UTC 29 January 2018 (extratropical trade wind regime) and (b) at 12 UTC 2 February 2018 (extratropical dry intrusion regime). The skew-T-log-p diagrams show temperature (brown lines) and dew point temperature (blue lines). The measured soundings are shown by thick lines, pseudo soundings from ERA5 data are shown by the thin lines for comparison. The measured horizontal wind is shown by the wind barbs on the right-hand side of the diagrams in black and the winds from the pseudo sounding are shown by grey wind barbs.**


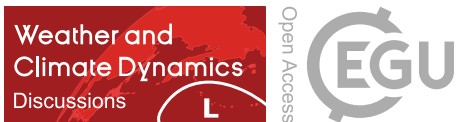

**Figure 10: Cloud patterns (a,c) over the North Atlantic and (b,d, 48-60°W, 10-20°N) around Barbados at 1430 UTC 29 January 2018**
**(a,b) and 1430 UTC 02 February 2018 (c,d) as seen from the MODIS instrument on TERRA. Images from NASA Worldview**
**Snapshots (https://wvs.earthdata.nasa.gov). In (a) the cyclone is visible that is associated with the extratropical dry intrusion**
**reaching Barbados a few days later on 1-4 February 2018.**



**Figure 11: Back-trajectories from the BCO started at (a) 15 UTC 29 January 2018 (extratropical trade wind regime) and (b) 15 UTC 02 February 2018 (extratropical dry intrusion regime). The trajectories are coloured according to their pressure. The position 4 days before arrival above Barbados is indicated by a dot coloured with their arrival pressure at the BCO (blue in the sub-cloud layer, green in the cloud layer, orange in the free troposphere and red in the upper troposphere). The black cross indicates the location of Barbados (BCO). The grey contours show the sea level pressure and the red contour denotes 2 pvu on 320 K, 4 days before arrival.**



**Figure 12: Upper-level PV on 320 K at 15 UTC on 20 (a) and the 26 January 2018 (b). During this period, air parcels that reach Barbados between 28-30 January descend above Portugal, Spain, and North Africa (position of air parcels shown by dots). The air parcels that experience rapid descent (>300 hPa(2d)$^{-1}$) centred at the time shown in the figure are highlighted in red (5 air parcels in a). The red cross indicates the position of Barbados. An animation illustrating the 3 hourly evolution of the large-scale flow fields in the period 20$^{th}$ to 30$^{th}$ of January 2018 is provided as Supplementary material S2.**

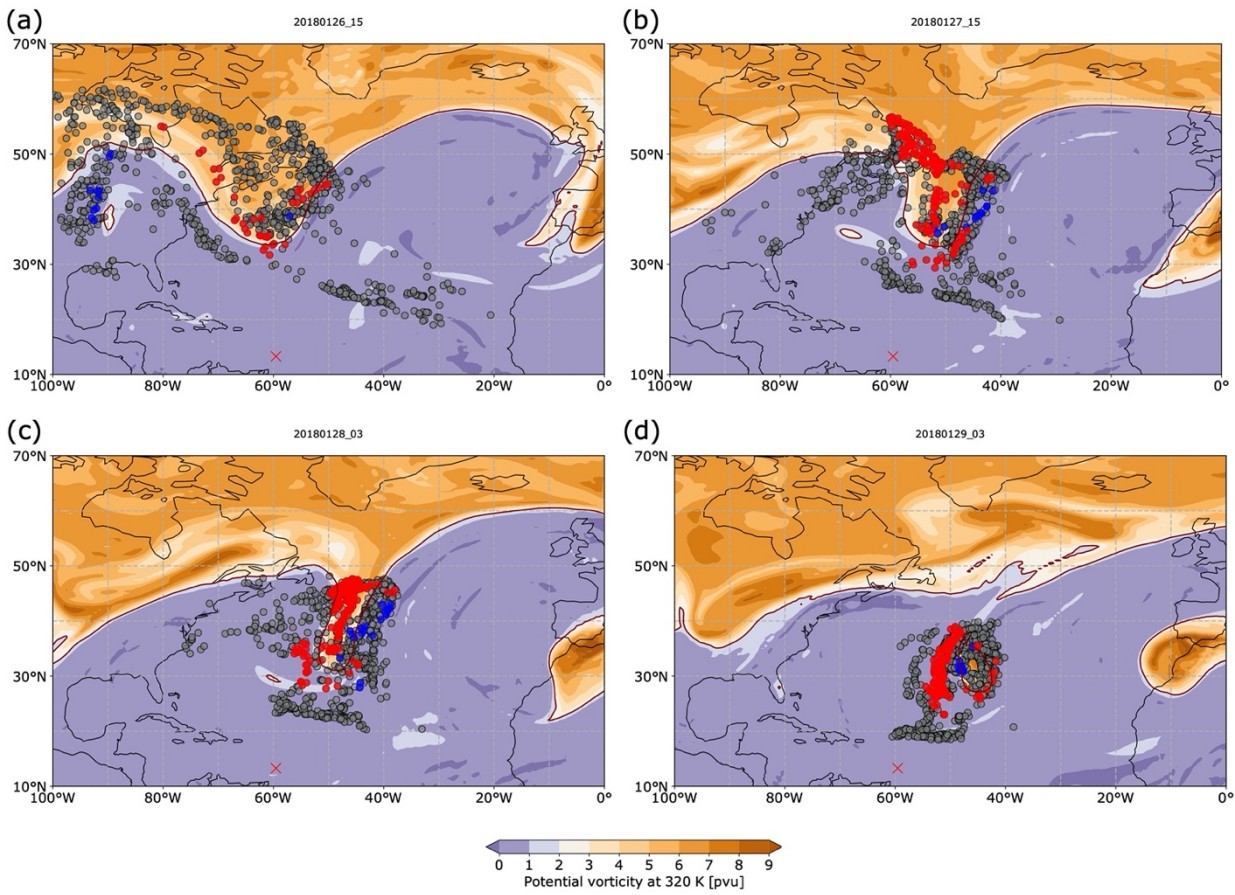

**Figure 13: The same as Fig. 12 but from 26-29 January 2018, and with air parcels that reach Barbados between 1-4 February. The air parcels that experience rapid ascent (<-300 hPa(2d)$^{-1}$) or descent (>300 hPa(2d)$^{-1}$) centred at the time shown in the figure are highlighted in blue respectively red. An animation illustrating the 3 hourly evolution of the large-scale flow fields in the period 26 January to 4 February 2018 is provided as Supplementary material S3.**



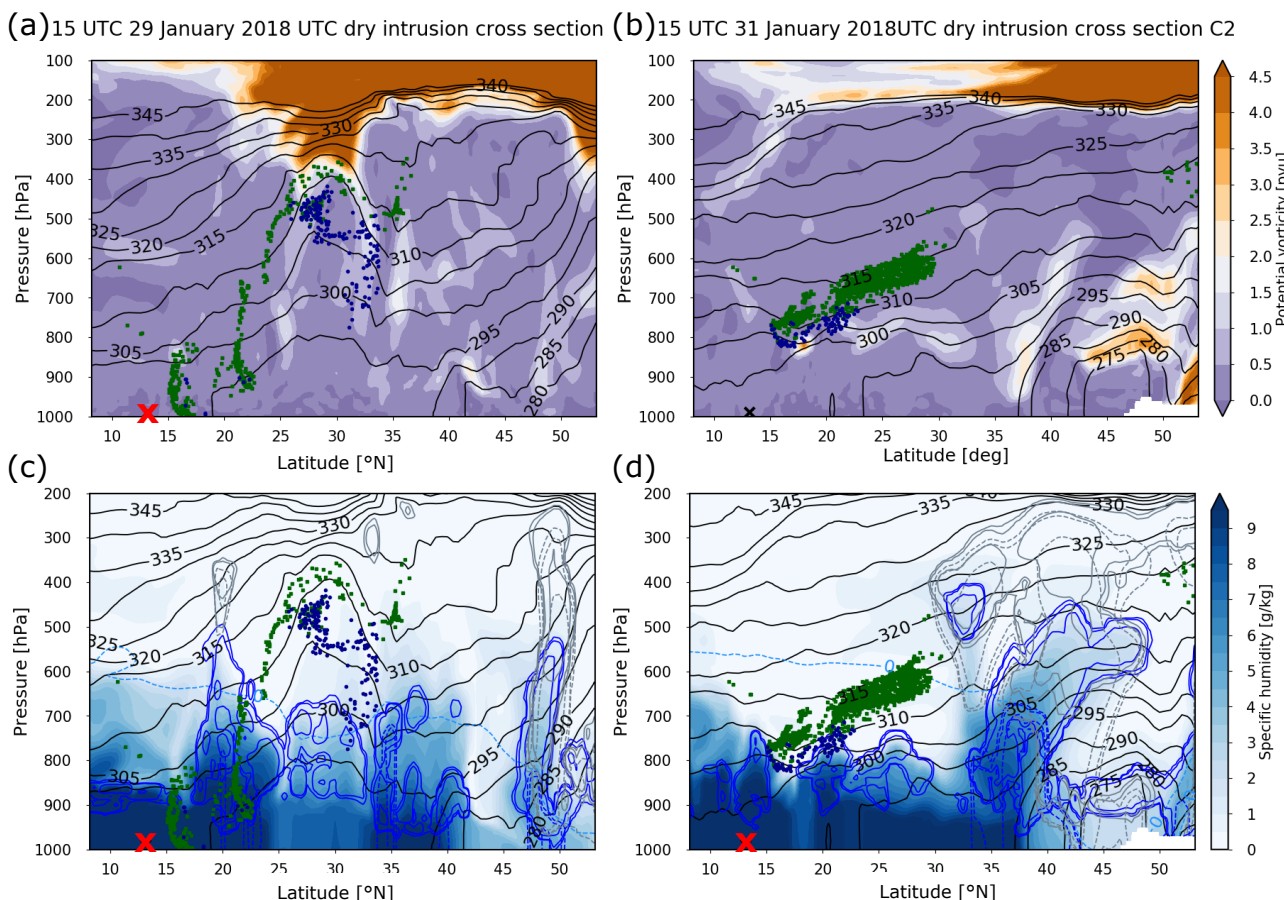

**Figure 14: Meridional cross sections along the lines C1 and C2 in Fig. 11b at 15 UTC 29 January 2018 through the upper-level cutoff (C1, a,c) and at 15 UTC 31 January 2018 showing the strongly descending air parcels west of the cutoff (C2, b,d). Filled contours show (a,b) potential vorticity and (c,d) specific humidity. Isentropes are shown by black contours, every 5 K. In (c,d) cloud liquid water is shown by solid blue contours, rain water by dashed blue, ice water content by grey, and snow water content by dashed grey contours. All hydrometeor contour lines correspond to 5, 10 and 50 mg kg$^{-1}$. The green and blue points correspond to the positions of air parcels within ± 1 ° longitude of the cross section. The green air parcels form together the cloud layer (940-700 hPa) and the blue air parcels form the sub-cloud layer (surface to 940hPa) above Barbados during the period of the extratropical dry intrusion between 1-4 February 2018. The latitude of Barbados is highlighted by the red cross at the surface.**





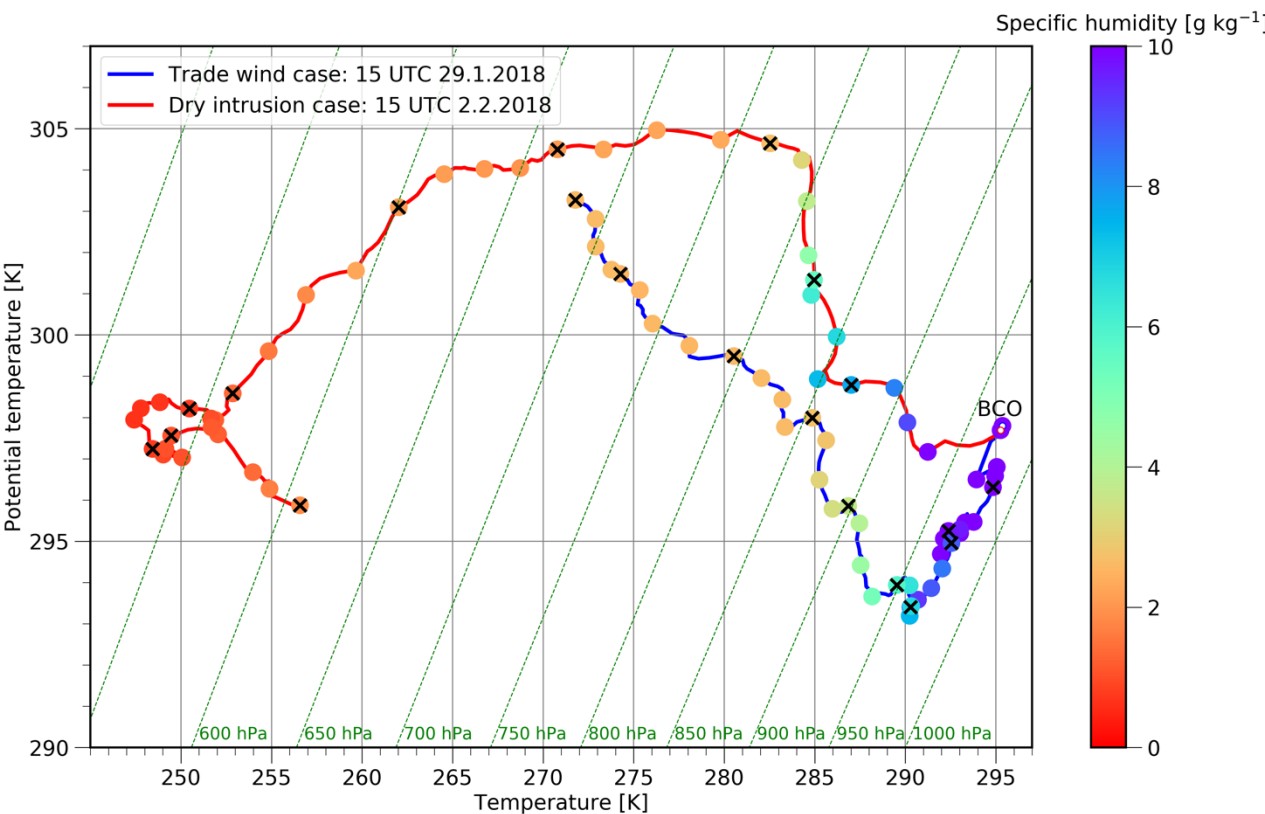

**Figure 15: Thermodynamic θ-T diagram summarising the 10-day Lagrangian history of the air parcels that arrived in the sub-cloud layer ($p > 940$ hPa) above Barbados at 15 UTC 29 January 2018 (blue line, extratropical trade wind regime) and at 15 UTC 2 February 2018 (red line, extratropical dry intrusion regime). Values are shown every 6 hours and averaged over the 40 sub-cloud layer trajectories. The colours of the filled circles indicate the specific humidity of the air parcels every 6 hours. The arrival**
**conditions above Barbados are indicated by white dots, the black crosses indicate daily time steps backwards in time. The slanted green lines show isobars, the horizontal lines isentropes and the vertical lines isotherms.**



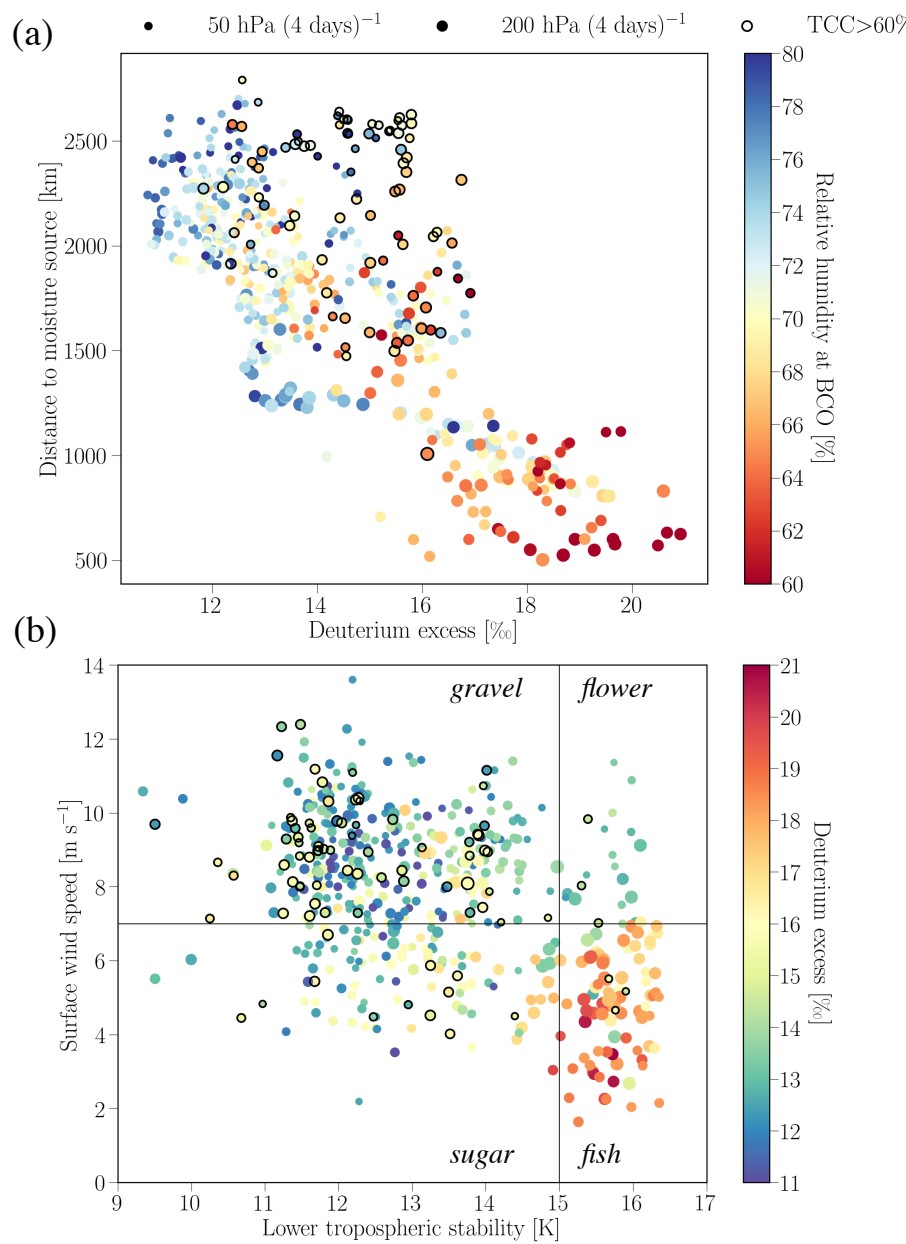

**Figure 16: Summary scatter plots showing in (a) the relation between the *d* with the moisture source distance (pearson correlation -0.73), and in (b) the relation between the *d* and the variables that allow to discriminate between different cloud patterns (the surface wind speed and the lower tropospheric stability, see Bony et al., 2020b) The pearson correlation between the *d* and the lower tropospheric stability *r(d,LTS)*=0.52 and between the *d* and the surface wind speed *r(d,U)*=-0.57. In both (a) and (b), the size of the dots is scaled with Δ*p*₄d the subsidence rate within 4 days along backward trajectories started from Barbados. The data points with total cloud cover of more than 60% along the trajectories are circled in black.**





**Table 1: Occurrence frequency of the three flow regimes defined in this study based on a classification of all hourly time steps in January-February in the period 2009 to 2018. The tropical flow regime is defined with $\tilde{\tau}_{35} < 1$ day. This regime also includes air parcels that descend from the subtropics. The share of purely tropical air parcels with $\tilde{\tau}_{23.5} < 1$ day is shown in parentheses. The extratropical trade wind regime is defined with with $\tilde{\tau}_{35} \geq 1$ day and a limited subsidence in the last 4 days prior to arrival in Barbados (< 100 hPa (4d)$^{-1}$); and extratropical dry intrusions with a residence time of at least one day in the extratropics and enhanced subsidence ($\geq$ 100 hPa (4d)$^{-1}$).**

| Dataset | Tropical | Extratropical trade wind | Extratropical dry intrusion |
|---|---|---|---|
| Jan-Feb 2009-2018 | 44% (11%) | 28% | 27% |
| Jan-Feb 2018 (isoTrades) | 6% (0%) | 62% | 32% |





**Table 2:** Lagrangian characteristics of backward trajectories from the BCO classified in either the extratropical trade wind or dry intrusion regime. See text for details. $\Delta p_{4d} = p(t_{BCO}) - p(t_{BCO} - 4 \text{ days})$ is the slantwise subsidence rate, $\max(\Delta p_{2d})$ stands for the maximum subsidence within a 2 day period along each of the 10-day back-trajectories (see Section 2.1 and Appendix A). $\lambda$ is the latitude, $\phi$ the longitude, $RH$ the relative humidity, $q$ the specific humidity, $\phi_{msd}$ weighted mean moisture source longitude and $L_{msd}$ the distance to the centre of mass of the moisture source region. Mean conditions (±their standard deviations) for the isoTrades period are indicated in the table based on all the trajectories arriving in the sub-cloud layer at the BCO for the two regimes.

| Air parcel properties | Extratropical trade wind regime | Extratropical dry intrusion regime |
|---|---|---|
| $\Delta p_{4d}$ | (32±36) hPa | (194 ± 71) hPa |
| $\max(\Delta p_{2d})$ | (403±79) hPa (2 d)$^{-1}$ | (440±77) hPa (2 d)$^{-1}$ |
| $\lambda$ [°N], $\phi$ [°W] before $\max(\Delta p_{2d})$ | (47±9)°N, (35±27)°W | (41±11)°N, (51±23)°W |
| $\lambda$ [°N], $\phi$ [°W] after $\max(\Delta p_{2d})$ | (35±10)°N, (20±14)°W | (29±10)°N, (43±15)°W |
| Time (days before arrival at the BCO) at the end of $\max(\Delta p_{2d})$ | (6±2) days | (4±2) days |
| Percentage of time steps with at least 1 trajectory with $\max(\Delta p_{2d}) > 400$ hPa (2d)$^{-1}$ | 53% | 72% |
| $RH$ after $\max(\Delta p_{2d})$ | (53±30)% | (55±33)% |
| $q$ before $\max(\Delta p_{2d})$ | (1.4±1.5) g kg$^{-1}$ | (1.3±1.9) g kg$^{-1}$ |
| $q$ after $\max(\Delta p_{2d})$ | (3.8±2.5) g kg$^{-1}$ | (4.9±3.8) g kg$^{-1}$ |
| Percental moisture uptake during $\max(\Delta p_{2d})$ | (17±15)% | (27±22)% |
| Percental moisture uptake during $\max(\Delta p_{2d})$ selecting trajectories with $\max(\Delta p_{2d}) > 400$ hPa (2d)$^{-1}$ | (25±15)% | (36±20)% |
| Percental moisture uptake after $\max(\Delta p_{2d})$ | (73±18)% | (63±29)% |
| $\phi_{msd}$ | (42±3)°W | (48 ± 6)°W |
| $L_{msd}$ | (2028±354) km | (1348±563) km |





**Table 3: Campaign mean local conditions at the BCO as well as conditions associated with the extratropical trade wind and dry intrusion flow regimes, respectively, based on measurements (first 9 variables) and ERA5 data. The *p*-value of a Wilcoxon rank-sum test for the significance of the difference between the two regime's distributions is <0.01 except for *T*. *q* is the specific humidity, *T* the air temperature, *P* the rain rate, $n_{cp}$ the number of cold pools within a 12 hour time period centred at the respective hourly time steps, *U* is the wind speed, *D* the wind direction and *RH* the relative humidity measured at BCO. The evaporation rate (*E*), the total column water (*TCW*), the total cloud cover (*TCC*) and the lower tropospheric stability (*LTS*=$\theta_{700hPa}$-$\theta_{1000hPa}$) are extracted from ERA5 and averaged over a 2°x2° box around BCO (13 - 15°N and 58 - 60°W) to provide regionally averaged values of these variables. Note that the results do not change substantially if the variables are interpolated to the position of the BCO.**

| Conditions | Overall isoTrades | Extratropical trade wind regime | Extratropical dry intrusion |
|---|---|---|---|
| $q$ [g kg$^{-1}$] | 15.1±1.0 | 15.4±0.8 | 14.5±1.1 |
| $\delta^{18}$O [‰] | −10.8±0.4 | −10.6±0.3 | −11.1±0.5 |
| $\delta^{2}$H [‰] | −71.8±1.9 | −71.5±1.8 | −72.5±2.1 |
| $d$ [‰] | 14.2±2.2 | 13.3±1.5 | 15.9±2.4 |
| $T$ [°C] | 26.0±0.6 | 26.0±0.7 | 26.0±0.5 |
| $n_{cp}$ [-] | 2.8±2.7 | 3.3±2.9 | 1.6±2.0 |
| $U$ [m s$^{-1}$] | 7.6±2.2 | 8.2±1.9 | 6.7±2.3 |
| $D$ [°N] | 75±13 | 78±9 | 68±17 |
| $RH$ [%] | 71±5 | 72±5 | 69±6 |
| $E$ [mm h$^{-1}$] | 0.31±0.05 | 0.31±0.05 | 0.33±0.05 |
| $TCW$ [mm] | 30±5 | 32±4 | 26±4 |
| $TCC$ [%] | 38±27 | 40±26 | 31±27 |
| $LTS$ [K] | 13.3±1.5 | 12.9±1.2 | 14.0±1.8 |





**Table 4: Isotope composition of precipitation events sampled with a totalisator from PALMEX. The samples were measured using a cavity ring-down laser spectrometer and normalised to the official IAEA VSMOW-SLAP scale. The analytical precision of the measurements is 0.16‰ for δ$^{18}$O and 0.6‰ for δ$^{2}$H.**

| Start date, time UTC | End date, time UTC | δ$^{18}$O (‰) | δ$^{2}$H (‰) | $d$ (‰) |
|---|---|---|---|---|
| 24 January 2018, 17:05 | 25 January 2018, 13:55 | –0.61 | 7.24 | 12.09 |
| 25 January 2018, 13:55 | 27 January 2018, 16:42 | –0.85 | 5.08 | 11.90 |
| 27 January 2018, 16:42 | 29 January 2018, 13:15 | –0.75 | 6.59 | 12.60 |
| 29 January 2018, 15:50 | 30 January 2018, 15:30 | 0.22 | 12.08 | 10.28 |
| 02 February 2018, 23:00 | 04 February 2018, 21:30 | –0.40 | 7.71 | 10.93 |
| 04 February 2018, 21:30 | 05 February 2018, 17:40 | 0.29 | 10.46 | 8.13 |
| 05 February 2018, 17:40 | 07 February 2018, 17:45 | –0.83 | 6.05 | 12.69 |
| 07 February 2018, 17:45 | 09 February 2018, 22:30 | 0.59 | 11.98 | 7.27 |
| 09 February 2018, 22:30 | 12 February 2018, 17:00 | –0.66 | 6.32 | 11.62 |
| 12 February 2018, 17:00 | 14 February 2018, 14:45 | –1.02 | 4.44 | 12.59 |
| 15 February 2018, 13:00 | 16 February 2018, 17:00 | –0.07 | 11.98 | 12.57 |
| 16 February 2018, 17:00 | 18 February 2018, 13:30 | –0.82 | 7.74 | 14.30 |
| Campaign average | | –0.41±0.53 | 8.14±2.76 | 11.41±2.0 |