# Peer review of "How Rossby wave breaking modulates the water cycle in the North Atlantic trade wind region"

_Weather and Climate Dynamics, 2020_

## Referee Comment (RC1) · Anonymous Referee #1 · 9 Nov 2020

General Comments

The manuscript addresses important questions regarding the influence of Rossby wave breaking on precipitation events that take place in the tropical North Atlantic. The questions belong to the more general framework of better understanding the interaction between low-level clouds in tropical regions and the large-scale circulation, which remains an outstanding problem in our current understanding of the climate system. The work presented here is based on a series of analyses conducted using a combination of reanalysis and a Lagrangian model, as well as data collected during a 24-day observational campaign that took place in Barbados in 2018. Notably, the data include

water vapor and liquid water isotopes. The authors show that Rossby wave breaking can play an important role in the water cycle at Barbados, and they identify and analyze patterns and mechanisms through which this can happen. I think that the paper addresses important questions with a sound methodology. The findings are very relevant, particularly as they show how high-resolution collection of water isotopes can provide useful information to advance our understanding of the climate system. I recommend the paper to be published after the authors have addressed a series of minor comments that are outlined below.

Specific Comments

- I understand that during the observational period, the "tropical flow regime" did not occur with a high frequency. However, in principle this does not imply that this flow regime could not occur with a higher frequency at other times. Would you consider including some discussion about that instead of neglecting it?

- Lin 138: This way of defining residence time does not seem to distinguish between parcels that are above a certain latitude continuously or in discontinuous intervals (provided that the totals are the same, obviously). Does this have an impact on your study?

- Judging by Figure 3, it would appear that 2018 is anomalous with respect to the climatology analyzed. Could you provide some explanation of why that is, maybe connecting it to large-scale modes of variability in the Atlantic?

- Line 407: The interpretation appears sound, but I do not understand the mechanism you are proposing. Cold pool gust fronts tend to be separated by the rain shafts of the clouds that generate them. How can they be influencing the re-evaporation of rain drops? Did you mean cold pools from another cloud?

- I found the fact that cloud patterns are associated with different deuterium excess very fascinating, although this is very briefly discussed in the manuscript. Would you consider expanding the discussion around Figure 16 (b) a little?

Technical Comments

- Line 146: Do I see correctly that Figure 3 is referenced before Figure 2? In this case, I would recommend switching the order.

- Figure 11, 12, 13: Please include axis labels

- Could you please ensure that the figures are color-blind friendly?

---

## Referee Comment (RC2) · Anonymous Referee #2 · 14 Nov 2020

In this manuscript, the authors present a detailed trajectory study of air parcels reaching Barbados and argue for the importance of extratropical processes in controlling the atmospheric hydrologic cycle in the North Atlantic trade region. It's a timely paper, coming soon after the EUREC4A campaign, and quite relevant to several ongoing issues involving the links between clouds, circulation, and climate. I found the paper interesting and well-written. I have no major methodological concerns, but I have several questions and comments that I hope will strengthen the presentation and clarify some of the arguments the authors are making.

(1) What's the benefit of the isotopes to this study? Other than identifying some basic

consistency between the back-trajectories and the isotopic measurements, what are we learning from the isotopes? In this setting, the variability in water vapor isotopic composition is very small and co-varies with the relative and specific humidities. so what's the added value of the isotopes? Figure 16 is an interesting part of the paper, but isn't discussed enough. Would you be able to generate a similar figure just using specific humidity instead of d-excess, or is the d-excess actually telling us something that we couldn't determine from other humidity fields? Given the expanding interest in water vapor isotopes in climate studies, this seems like a missed opportunity to 'sell' the audience on the value of such measurements.

(2) I am confused by the calculations involved in the moisture uptake analysis. When the authors say that moisture sources are located, for example, 2000 km away from Barbados, does this mean that the actual molecules of water that are measured in their Picarro were evaporated from the ocean surface that far away? That seems like a surprising result - I would certainly think that the vast majority of water molecules measured at BCO would have been evaporated from the sea surface not far offshore. If the water vapor is really evaporated so far away, would the isotopic composition of that water vapor reflect the SSTs that far upstream? Is that the case, or are the isotopic measurements mostly consistent with SSTs in the immediate vicinity? Or perhaps the SSTs don't vary enough along the trajectory to generate a measurable difference? Does the Sodemann model predict the isotopic composition of the water vapor? It seems that this would be an opportunity for the isotopes to provide some real-world constraints on a model result. In any case, some discussion on this point would be worthwhile.

(3) The authors refer to 'air parcels' and quantify frequencies of occurrence, but isn't the definition of air parcels here based on the frequency with which they are running the back trajectories? It's a minor point, I suppose, but doesn't that affect the frequency calculations? Would you get different percentages if the trajectories were run every 15 minutes or every 3 hours?

---

## Author Comment (AC1) · 15 Jan 2021

Please find our response to the reviewers' comments attached.

Please also note the supplement to this comment:
https://wcd.copernicus.org/preprints/wcd-2020-51/wcd-2020-51-AC1-supplement.pdf

---

## Author Response (AR1)

**Response to the reviewers' comments**

We thank the two reviewers for their insightful comments. Below, we address each comment point by point. The reviewers' comments are repeated in blue, our responses are given in black and the changes to the manuscript in black italic.

**Reviewer 1**

**General Comments**

The manuscript addresses important questions regarding the influence of Rossby wave breaking on precipitation events that take place in the tropical North Atlantic. The questions belong to the more general framework of better understanding the interaction between low-level clouds in tropical regions and the large-scale circulation, which remains an outstanding problem in our current understanding of the climate system. The work presented here is based on a series of analyses conducted using a combination of reanalysis and a Lagrangian model, as well as data collected during a 24-day observational campaign that took place in Barbados in 2018. Notably, the data include water vapor and liquid water isotopes. The authors show that Rossby wave breaking can play an important role in the water cycle at Barbados, and they identify and analyze patterns and mechanisms through which this can happen. I think that the paper addresses important questions with a sound methodology. The findings are very relevant, particularly as they show how high-resolution collection of water isotopes can provide useful information to advance our understanding of the climate system. I recommend the paper to be published after the authors have addressed a series of minor comments that are outlined below.

We thank the reviewer for this positive general feedback.

**Specific Comments**

1) I understand that during the observational period, the "tropical flow regime" did not occur with a high frequency. However, in principle this does not imply that this flow regime could not occur with a higher frequency at other times. Would you consider including some discussion about that instead of neglecting it?

In this paper, we focused on the role of the two extratropical flow regimes, because these regimes were dominant during the isoTrades campaign and because we think that they are often overlooked when the circulation in the trades during winter is discussed. We do however not want to imply that the tropical flow regime is overall rare. In a follow-up study on the flow regimes observed during the EUREC$^4$A campaign in Jan-Feb 2020 we will discuss the tropical flow regime in more detail. The latter regime was much more prominent during EUREC$^4$A in 2020 than during isoTrades in 2018, which is why we prefer to keep the discussion about this regime very short in this study.

For the reviewer's own interest, we show in Fig. 1 in this document the moisture source composite for the tropical flow regime during isoTrades. Compared to the two other flow regimes (see Fig. 6 in the manuscript), the moisture source pattern is much more zonal, with some influence of air parcels taking up humidity in the subtropics during their descent towards the trades.

[Figure]

Figure 1: Moisture sources of the tropical flow regime during isoTrades (in filled contours). Other variables shown as in Fig. 6 in the manuscript.

2) Line 138: This way of defining residence time does not seem to distinguish between parcels that are above a certain latitude continuously or in discontinuous intervals (provided that the totals are the same, obviously). Does this have an impact on your study?

Thank you for this remark, it is indeed true that in some instances, the air parcels originating from the midlatitude jet stream region travel zonally within the lower part of the jet along the edge of the subtropics. We therefore tested the influence of our residence time definition by computing the maximum continuous residence time in the extratropics $\tau_{35,\text{max}}$ for each trajectory. We compared $\tau_{35,\text{max}}$ to the total residence time in the extratropics $\tau_{35}$ for the year 2018 and found that the two were equal for 91% of the trajectories arriving in the sub-cloud layer. For trajectories arriving above the sub-cloud layer this value reduces to 80%. Thus, the oscillating behaviour of air parcels around the subtropical boundary, when travelling zonally within the jet is slightly more frequent for air parcels arriving in the cloud layer and in the free troposphere above Barbados ($p$ < 940 hPa). We therefore conclude that our simple definition of the residence time is overall meaningful and that considering continuous or discontinuous intervals of air parcel residence north of 35°N only has a very small impact on the calculated residence time climatology.

3) Judging by Figure 3, it would appear that 2018 is anomalous with respect to the climatology analyzed. Could you provide some explanation of why that is, maybe connecting it to large-scale modes of variability in the Atlantic?

When submitting the manuscript, we have decided on purpose to not discuss the anomalies of the year 2018 with respect to the North Atlantic circulation characteristics in order to keep this study focused. But since this question apparently arises in the reviewers' comments, we decided to add a short paragraph about the link between the North Atlantic Oscillation and anticyclonic Rossby wave breaking. We now discuss this aspect in section 3.1, at line 280 focussing on its possible contribution in the trade wind flow anomaly observed in 2018.

*p.9, L278: "The isoTrades campaign period was characterised by an unusually large influence of air parcels originating from relatively far North compared to the climatology. This anomalous behaviour is linked to a poleward shift of the midlatitude jet in January-February*

*2018 over the western North Atlantic, and a positive North Atlantic Oscillation index (NAO, see Supplement S2). Several studies have shown that ARWB over the North Atlantic occurs preferentially in the positive phase of the NAO (Benedict et al., 2004; Rivière and Orlanski, 2007). Furthermore, January-February 2018 was characterised by a relatively weak deep convective activity in the northern part of South America as well as a southward shift of the ITCZ over the South Atlantic compared to climatology (see Supplement S2). Both the northward shift of the midlatitude jet and the anomalies of the ITCZ may have played a role in the anomalously strong contribution of air parcels from the extratropics during isoTrades."*

As expected from the above described relation between ARWB over the North Atlantic and the NAO, we found a weak positive correlation (Pearson correlation 0.37) between the NAO and the percentage of air parcels originating from the extratropics over the climatological period January-February 2009-2018. However, we think that for the share of extratropical air parcels in the sub-cloud layer in Barbados, other dynamical aspects that are not directly reflected by the NAO index are equally important such as the strength of the jet and the persistence of weak upper-level cutoff lows over the central North Atlantic favouring the direct transport pathway from the North towards the island. In the paper, we would therefore not like to expand the discussion about this aspect beyond the paragraph above.

4) Line 407: The interpretation appears sound, but I do not understand the mechanism you are proposing. Cold pool gust fronts tend to be separated by the rain shafts of the clouds that generate them. How can they be influencing the re-evaporation of rain drops? Did you mean cold pools from another cloud?

We mean during the initiation of precipitation-induced downdrafts of the thereafter developing cold pools, but also rain evaporation in the context of secondary convection generated by the cold pool gust fronts. We changed the text as follows:

*p. 13,* at L. 419*: "In precipitation-induced downdrafts or at the edge of cold pools, where secondary convection is generated, rapid total re-evaporation of small rain droplets (i.e. no net fractionation) may thus have contributed to the positive anomalies in $\delta$ values and negative anomalies in d observed during the trade wind regime periods of the campaign."*

5) I found the fact that cloud patterns are associated with different deuterium excess very fascinating, although this is very briefly discussed in the manuscript. Would you consider expanding the discussion around Figure 16 (b) a little?

Thank you for this comment. We agree and expanded the discussion accordingly. We added the following two paragraphs in Section 3.5:

*p. 17, L. 529: "The high d anomalies during the extratropical dry intrusions are associated with the Fish cloud pattern, while the low d anomalies in the trade wind flow regime are associated with the Gravel cloud pattern (Fig. 16b). These characteristic d fingerprints in the Fish and Gravel cloud patterns suggest important differences in the water vapour cycling for these two patterns. Based on the isoTrades campaign data, the Fish cloud pattern with anomalously low surface wind speeds and high LTS seems to be associated with extratropical dry intrusions behind trailing cold front passages. The calm conditions in this flow regime are likely due to the interruption of the normally near-zonal trade wind flow by the southward moving air. The high LTS might be due to the cold advection in the sub-cloud layer and the enhanced subsidence aloft. The potential link between enhanced stability and the dry intrusion has to be investigated in more detail in a future study, by analysing the heat budget along backward*

*trajectories. The anomalously low d associated with the Gravel cloud pattern is due to the cloud processing and the in general more remote moisture uptake as discussed above. The influence of cold pool passages on the short-term variability of isotope signals in the trade wind regime, specifically during gust front passages, seems to be important. However, this short-term variability has a smaller amplitude than the high d anomalies produced at the synoptic timescale during the extratropical dry intrusion flow regime. The anomalous d signal during the dry intrusion regime thus provides observational evidence of the altered large-scale flow conditions compared to the trade wind regime and to some extent validates our findings based on the back-trajectory analysis.*

*Given the influence of low-cloud amount and potentially cloud organisation on cloud-radiative feedbacks (Bony et al., 2020b), the sensitivity of the cloud patterns to the different large-scale flow regimes in the trades has to be studied in more detail. In particular, the frequency of occurrence and persistence of the four cloud types identified in the surrounding of Barbados are likely tied to the prevailing flow regimes. This provides a promising starting point for a more detailed analysis of the link between different flow regimes and cloud patterns. In particular, the importance of the extratropical origin of air parcels during prolonged episodes with the Fish cloud pattern will be further investigated using the EUREC$^4$A isotope datasets in a follow-up study."*

Technical Comments

Line 146: Do I see correctly that Figure 3 is referenced before Figure 2? In this case, I would recommend switching the order.

This is true, we will change the order of these two Figures.

Figure 11, 12, 13: Please include axis labels

We added °N and °W/°E labels in Figure 11.

Could you please ensure that the figures are color-blind friendly?

We understand the request made by the reviewer and are ready to change any potentially confusing figure. However, we have checked all figures with a pdf viewer simulating different colour-blindness types and could not identify problematic figures. We would therefore be grateful if the reviewer could specify which figure and colours are meant here.

**Reviewer 2**

In this manuscript, the authors present a detailed trajectory study of air parcels reaching Barbados and argue for the importance of extratropical processes in controlling the atmospheric hydrologic cycle in the North Atlantic trade region. It's a timely paper, coming soon after the EUREC[4]A campaign, and quite relevant to several ongoing issues involving the links between clouds, circulation, and climate. I found the paper interesting and well-written. I have no major methodological concerns, but I have several questions and comments that I hope will strengthen the presentation and clarify some of the arguments the authors are making.

We thank the reviewer for this positive general feedback.

1) What's the benefit of the isotopes to this study? Other than identifying some basic consistency between the back-trajectories and the isotopic measurements, what are we learning from the isotopes? In this setting, the variability in water vapor isotopic composition is very small and co-varies with the relative and specific humidities. So what's the added value of the isotopes? Figure 16 is an interesting part of the paper, but isn't discussed enough. Would you be able to generate a similar figure just using specific humidity instead of d-excess, or is the d-excess actually telling us something that we couldn't determine from other humidity fields? Given the expanding interest in water vapor isotopes in climate studies, this seems like a missed opportunity to 'sell' the audience on the value of such measurements.

Thank you for this useful comment, we agree that we should discuss the added value of isotopes as tracers for moist atmospheric processes in a more explicit and prominent way in the paper. Below, we first provide more explanations of why isotopes benefit this study. In particular, we show how isotopes provide additional information compared to the traditional humidity measures. Second, we mention the changes made in the manuscript to make the key role of isotopes as tracers for moisture source and cloud processes clearer and more convincing.

The isotope signals indeed generally covary with locally measured traditional humidity measures such as specific humidity $q$ and relative humidity $RH$ (see Table 1 below).

Table 1: Isotope variables and their overall Pearson correlation of hourly data during isoTrades with traditional humidity variables (specific humidity q and relative humidity RH).

| Isotope variable | $q$ | $RH$ |
|---|---|---|
| $\delta^{18}O$ | 0.83 | 0.64 |
| $d$ | -0.47 | -0.56 |

A more detailed analysis shows that these covariations are subject to important variability in space and time and in particular are dependent on the synoptic situation (see Figs. 2 and 3 below and Aemisegger, 2013, Chapter 4; Aemisegger et al., 2014). For example, the 5-day moving window correlation between $\delta^{18}O$ and $q$ at a pre-alpine station in Switzerland was found to be generally positive as expected from the "distillation" effect or temperature effect described by Dansgaard in 1964, which is widely used in ice core paleo-climate reconstructions. This relation simply describes the progressive depletion of heavy isotopes in an air parcel as the air parcel is moist-adiabatically cooled and loses humidity by cloud formation and rain out. However, in our previous work (see references above) we observed a negative relation between $\delta^{18}O$ and $q$ during individual prolonged periods with moderate rainfall from stratiform cold sector clouds. We interpreted this negative relation as an effect

of progressive decrease in $\delta^{18}O$ of water vapour (by exchange with falling hydrometeors), while the specific humidity increases due to moistening of the below cloud air. A modelling study testing the sensitivity of simulated isotope signals with respect to below cloud and surface evaporation processes confirmed this interpretation and showed that the $\delta$-values and $d$ were very sensitive to below cloud effects during cold front passages over central Europe (Aemisegger et al., 2015). During isoTrades, the trailing cold front passage on 31 January also leads to a momentary decrease in the 24 h moving window $q$-$\delta^{18}O$ correlation (see Fig. 2 below). Furthermore, the anticorrelation between local $RH$ and $d$ is strong during the extratropical dry intrusion from 1 to 4 February but weak during the trade wind regime period of 29 to 30 January. Based on these more detailed analyses of the temporal correlation patterns between isotopes and the traditional humidity measures we know that isotopes provide additional information on the influence of microphysical processes related to clouds and falling hydrometeors.

[Figure]

Figure 2: Moving window (24 h, centred) Pearson correlation between q and d$^{18}$O in blue and q and d in red based on hourly data.

[Figure]

Figure 3: Moving window (24 h, centred) Pearson correlation between RH and d$^{18}$O in blue and RH and d in red based on hourly data.

In addition to the information gained from isotopes on cloud and below cloud processes, the relation between $d$ and the distance to the moisture source as well as $d$ and the subsidence rate in the last 4 days before arrival in Barbados is much stronger than for $q$ and $RH$ (Table 2 below). This is due to the fact that $q$ and $RH$ are not particularly sensitive to remote strong ocean evaporation events, while $d$ is (Aemisegger and Sjolte, 2018). Therefore, Fig. 16 from the manuscript adapted for $q$ shows a less clear relation between $q$ and the cloud patterns (Fig. 4 below) than between $d$ and the cloud patterns (Fig. 16 in the manuscript). Again, this

highlights the added information of stable water isotopes compared to the traditional humidity measures.

Table 2: Pearson correlations between different humidity measures with moisture source and transport *properties.*

| Humidity measure | Distance to the moisture source | Subsidence in the last 4 days |
|---|---|---|
| *d* | -0.73 | 0.58 |
| *q* | 0.18 | -0.19 |
| *RH* | 0.50 | -0.32 |

[Figure]

Figure 4: Left panel as Fig. 16 in the manuscript but for specific humidity: Relation between measured specific humidity at the BCO and cloud patterns. The right panel corresponds to Fig. 16 in the manuscript.

We have thus made the following changes to the manuscript:

a)  Introduction of isotope measurements in the results section:

p.13, L397*: "The stable water isotope signals reflect the contrasts in atmospheric water vapour cycling in the two synoptic flow situations. In particular, they provide information about moisture source and transport conditions, which traditional humidity variables do not reveal to the same extent. The physical foundations of the interpretation of the isotope signals have been published in earlier work and are only shortly mentioned and referenced here."*

b)  More detailed discussion of Fig. 16 on p. 17 (see also comment 5 from reviewer 1):

*p. 17, L. 529: "The high d anomalies during the extratropical dry intrusions are associated with the Fish cloud pattern, while the low d anomalies in the trade wind flow regime are associated with the Gravel cloud pattern (Fig. 16b). These characteristic d fingerprints in the Fish and Gravel cloud patterns suggest important differences in the water vapour cycling for these two patterns. Based on the isoTrades campaign data, the Fish cloud pattern with anomalously low surface wind speeds and high LTS seems to be associated with extratropical dry intrusions behind trailing cold front passages. The calm conditions in this flow regime are likely due to the interruption of the normally near-zonal trade wind*

*flow by the southward moving air. The high LTS might be due to the cold advection in the sub-cloud layer and the enhanced subsidence aloft. The potential link between enhanced stability and the dry intrusion has to be investigated in more detail in a future study, by analysing the heat budget along backward trajectories. The anomalously low d associated with the Gravel cloud pattern is due to the cloud processing and the in general more remote moisture uptake as discussed above. The influence of cold pool passages on the short-term variability of isotope signals in the trade wind regime, specifically during gust front passages, seems to be important. However, this short-term variability has a smaller amplitude than the high d anomalies produced at the synoptic timescale during the extratropical dry intrusion flow regime. The anomalous d signal during the dry intrusion regime thus provides observational evidence of the altered large-scale flow conditions compared to the trade wind regime and to some extent validates our findings based on the back-trajectory analysis.*

*Given the influence of low-cloud amount and potentially cloud organisation on cloud-radiative feedbacks (Bony et al., 2020b), the sensitivity of the cloud patterns to the different large-scale flow regimes in the trades has to be studied in more detail. In particular, the frequency of occurrence and persistence of the four cloud types identified in the surrounding of Barbados are likely tied to the prevailing flow regimes. This provides a promising starting point for a more detailed analysis of the link between different flow regimes and cloud patterns. In particular, the importance of the extratropical origin of air parcels during prolonged episodes with the Fish cloud pattern will be further investigated using the EUREC[4]A isotope datasets in a follow-up study."*

c) Covariation of specific humidity with *d* and short discussion about the added value of isotopes compared to traditional humidity measures:

   *p. 14, L. 423: "For a closer investigation of the isotope variability associated with cold pool passages in the extratropical trade wind regime, the isotope data should be used at a higher temporal resolution than hourly (e.g. 1 min), since the duration of cold pool passages is in the order of 20-30 min. Isotopes could provide more information about the source of the moist ring often observed at the edge of cold pools (Langhans and Romps, 2015; Torri and Kuang, 2016) and in general about the moisture budget of cold pools."*

   *p. 16, L522: "A clear observational linkage between the extratropical transport pathways and the isotope signals from the BCO is thereby obtained, which is not available from traditional humidity measures. Note that the relations between RH or q and the distance to the moisture source are much weaker than for d, and no physical rationale exists for their dependence to the distance to the moisture uptake region."*

2) I am confused by the calculations involved in the moisture uptake analysis. When the authors say that moisture sources are located, for example, 2000 km away from Barbados, does this mean that the actual molecules of water that are measured in their Picarro were evaporated from the ocean surface that far away? That seems like a surprising result - I would certainly think that the vast majority of water molecules measured at BCO would have been evaporated from the sea surface not far offshore. If the water vapor is really evaporated so far away, would the isotopic composition of that water vapor reflect the SSTs that far upstream? Is that the case, or are the isotopic measurements mostly consistent with SSTs in the immediate vicinity? Or perhaps the SSTs don't vary enough along the trajectory to generate a measurable difference? Does the Sodemann model predict the isotopic composition of the water vapor? It seems that this would be an opportunity for the isotopes to provide some real-world constraints on a model result. In any case, some discussion on this point would be worthwhile.

We are happy to provide more information about the assumptions involved in the Lagrangian moisture source diagnostics, which is based on the algorithm introduced by Sodemann et al. (2008a). This method has been extensively used in the past in the context of water isotope studies (see Sodemann et al., 2008b; Pfahl and Wernli, 2008, 2009; Aemisegger et al., 2014; Dütsch et al., 2018; Aemisegger, 2018; Thurnherr et al., 2020a,b) and for studies about the moisture sources of (heavy) precipitation (see Sodemann and Zubler, 2010; Pfahl et al., 2014; Winschall et al., 2014a,b; Läderach and Sodemann, 2016; Sodemann, 2020). From these studies we know that:

A) The measured deuterium excess generally correlates better with diagnosed moisture source conditions than with local conditions in particular during periods when large-scale transport dominates. This specifically shows the added value of water vapour and precipitation isotope measurements for validating the trajectory approach.

B) An overall consistent estimate of the geographical moisture source distribution and the lifetime of water vapour in the atmosphere is obtained from studies applying the Lagrangian moisture source approach used here and an Eulerian approach using diagnostic numerical tracers in high resolution regional numerical weather prediction models (see e.g. Sodemann et al., 2009; Winschall et al., 2014a).

Based on the current state of knowledge, the lifetime of water vapour in the atmosphere, meaning the actual time that water molecules spend in the atmosphere between evaporation and precipitation (Trenberth, 1998) is in the range of 2 to 10 days. First estimates of 8-10 days date back to the study by Peixóto and Oort (1983), who used radiosonde and gridded precipitation data. More recent Lagrangian studies based on reanalysis data provide estimates of 4 to 6 days in the median (Läderach and Sodemann, 2016; Sodemann, 2020). Note that the latter studies also emphasise that the water vapour lifetime distribution is skewed and has a long tail. A very simple order of magnitude calculation can be done by estimating the steady state replacement time of the entire water vapour of the atmospheric column in the trade wind region by ocean evaporation. With a total column water of 33 mm and an evaporation flux of 0.3-0.4 mm h$^{-1}$ this yields 3.5 to 4.5 days. This estimate is in agreement with our more sophisticated Lagrangian approach used in this paper, in which we explicitly consider the water vapour budget of a moving air parcel.

In our revised paper, the Supplementary material S1 provides more details on the moisture source diagnostic tool. We now mention S1 in the methods Section, where the trajectory-based diagnostics are presented:

p. 6 at L.185: *"More details about the applied moisture source diagnostics can be found in the Supplementary material S1."*

Two additional remarks with respect to the reviewer's comment:
- The deuterium excess reflects primarily the near-surface humidity gradient expressed generally as a relative humidity with respect to sea surface temperature ($RH_{SST} = \frac{e(T_d)}{e(SST)}$, with $e$ the saturation vapour pressure at the dew point temperature ($T_d$) and the sea surface temperature ($SST$)). The SST plays a secondary role in modulating atmospheric water vapour deuterium excess especially at the synoptic time-scale (see also Aemisegger and Sjolte, 2018).
- The method of Sodemann et al. (2018) is independent of isotopes and is not an isotope-enabled trajectory-based box model. Such models exist and have been used in the past, primarily in polar isotope studies and for the interpretation of snow data (e.g. Helsen et al., 2007). The use of such a model is however beyond the scope of this manuscript.

3) The authors refer to 'air parcels' and quantify frequencies of occurrence, but isn't the definition of air parcels here based on the frequency with which they are running the back trajectories? It's a minor point, I suppose, but doesn't that affect the frequency calculations? Would you get different percentages if the trajectories were run every 15 minutes or every 3 hours?

No, the frequency of occurrence of the regimes does not depend substantially on the frequency at which we are running the trajectories. The differences in the results when running the trajectories three-hourly are negligible and only due to the difference in sample size. Differences of ~1% in the occurrence frequencies of the defined flow regimes appear, when starting the back trajectories every day instead of every hour (see Table 3 below).

Our results however do depend slightly on the resolution of the reanalysis data used for the trajectory calculation. We additionally computed the trajectories during the isoTrades time period in 2018 with 3D wind fields from ERA-Interim reanalyses (Dee et al., 2011; 1° horizontal resolution, 60 vertical levels) with trajectories started every 6-hours, with 6-hourly output along the trajectories. Thereby we obtained the results shown in Table 3 below, which are similar to the ones presented in the paper. The differences in the occurrence frequencies obtained with ERA-Interim compared to ERA5 for the extratropical regimes come from the fact that the vertical winds are weaker in the ERA-Interim dataset. The chosen threshold of 100 hPa (4d)$^{-1}$ for the subsidence rate corresponds to the 65$^{th}$ percentile in ERA5 and the 75$^{th}$ percentile in ERA-Interim.

Table 3: Occurrence frequencies of the three studied regimes based on trajectories computed with ERA5 compared to ERA-Interim wind fields. Since the tropical flow regime is defined with $\tilde{\tau}_{35}$ < 1 day, this regime also includes air parcels that descend from the subtropics. Therefore, as in Table 1 of the manuscript, the share of purely tropical air parcels with $\tilde{\tau}_{23.5}$ < 1 day is shown in parentheses.

| Dataset Jan-Feb 2018 (isoTrades) | Tropical | Extratropical trade wind | Extratropical dry intrusion |
|---|---|---|---|
| hourly ERA5 trajectories (as in the paper) | 6% (0%) | 62% | 32% |
| 6-hourly ERA5 trajectories | 7% (0%) | 61% | 32% |
| 6-hourly ERA-Interim | 6% (0%) | 70% | 24% |

**References**

Aemisegger, F., Pfahl, S., Sodemann, H., Lehner, I., Seneviratne, S. I., and Wernli, H.: Deuterium excess as a proxy for continental moisture recycling and plant transpiration, Atmos. Chem. Phys., 14, 4029–4054, https://doi.org/10.5194/acp-14-4029-2014, 2014.

Aemisegger, F. and Sjolte, J.: A climatology of strong large-scale ocean evaporation events. Part II: Relevance for the deuterium excess signature of the evaporation flux, J. Climate, 31, 7313–7336, https://doi.org/10.1175/JCLI-D-17-0592.1, 2018.

Aemisegger, F., Spiegel, J. K., Pfahl, S., Sodemann, H., Eugster, W., and Wernli, H.: Isotope meteorology of cold front passages: A case study combining observations and modeling, Geophys. Res. Lett., 42, 5652–5660, https://doi.org/10.1002/2015GL063988, 2015.

Aemisegger, F., Sturm, P., Graf, P., Sodemann, H., Pfahl, S., Knohl, A., and Wernli, H.: Measuring variations of $\delta^{18}$O and $\delta^2$H in atmospheric water vapour using two commercial laser-based spectrometers: an instrument characterisation study, Atmos. Meas. Tech., 5, 1491–1511, https://doi.org/10.5194/amt-5-1491-2012, 2012.

Aemisegger, F.: On the link between the North Atlantic storm track and precipitation deuterium excess in Reykjavik, Atmos. Sci. Lett., 19, e865, https://doi.org/10.1002/asl.865, 2018.

Aemisegger, F.: Atmospheric stable water isotope measurements at the timescale of extratropical weather systems, Chapter 4, ETH Diss no 21165, https://doi.org/10.3929/ethz-a-009989698, 2013.

Benedict, J. J., Lee, S., and Feldstein, S. B.: Synoptic view of the North Atlantic Oscillation, J. Atmos. Sci., 61, 121–144, doi: 10.1175/1520-0469(2004)061<0121:SVOTNA>2.0.CO;2, 2004.

Dansgaard, W.: Stable isotopes in precipitation, Tellus, 16, 436–468, https://doi.org/10.3402/tellusa.v16i4.8993, 1964.

Dütsch, M., Pfahl, S., Meyer, M., and Wernli, H.: Lagrangian process attribution of isotopic variations in near-surface water vapour in a 30-year regional climate simulation over Europe, Atmos. Chem. Phys., 18, 1653–1669, https://doi.org/10.5194/acp-18-1653-2018, 2018.

Helsen, M. M., Van de Wal, R. S. W., and Van den Broeke, M. R.: The isotopic composition of present-day Antarctic snow in a Lagrangian atmospheric simulation, J. Climate, 20, 739–756, https://doi.org/10.1175/JCLI4027.1, 2007.

Langhans, W., and Romps, D. M.: The origin of water vapor rings in tropical oceanic cold pools, Geophys. Res. Lett., 42, 7825–7834, doi:10.1002/2015GL065623, 2015.

Läderach, A., and Sodemann, H.: A revised picture of the atmospheric moisture residence time, Geophys. Res. Lett., 43, 924–933, doi:10.1002/2015GL067449, 2016.

Peixóto, J., and Oort, A.: The atmospheric branch of the hydrological cycle and climate. Variations in the Global Water Budget, A. Street-Perrott, M. Beran, and R. Ratcliffe, Eds., Springer, 5–65, https://doi.org/10.1007/978-94-009-6954-4_2, 1983.

Pfahl, S. and Wernli, H.: Air parcel trajectory analysis of stable isotopes in water vapor in the eastern Mediterranean, J. Geophys. Res, 113, D20104, doi:10.1029/2008JD009839, 2008.

Pfahl, S. and Wernli, H.: Lagrangian simulations of stable isotopes in water vapor - an evaluation of non-equilibrium fractionation in the Craig-Gordon model. J. Geophys. Res., 114, D20108, doi:10.1029/2009JD012054, 2009.

Pfahl, S., Madonna, E., Boettcher, M., Joos, H., and Wernli, H.: Warm conveyor belts in the ERA-Interim data set (1979-2010). Part II: Moisture origin and relevance for precipitation. J. Climate, 27, 27-40, doi:10.1175/JCLI-D-13-00223.1, 2014.

Sodemann, H., Schwierz, C., and Wernli, H.: Interannual variability of Greenland winter precipitation sources: Lagrangian moisture diagnostic and North Atlantic Oscillation influence, J. Geophys. Res.-Atmos., 113, D03107, https://doi.org/10.1029/2007JD008503, 2008a.

Sodemann, H., Masson-Delmotte, V., Schwierz, C., Vinther, B. M., and Wernli, H.: Interannual variability of Greenland winter precipitation sources: 2. Effects of North Atlantic Oscillation variability on stable isotopes in precipitation, J. Geophys. Res., 113, D12111, https://doi.org/10.1029/2007JD009416, 2008b.

Sodemann, H.: Beyond turnover time: Constraining the lifetime distribution of water vapor from simple and complex approaches, J. Atmos. Sci., 77, 413–433, https://doi.org/10.1175/JAS-D-18-0336.1, 2020.

Sodemann, H. and Zubler, E.: Seasonal and inter-annual variability of the moisture sources for Alpine precipitation during 1995– 2002, Int. J. Climatol., 30, 947–961, doi:10.1002/joc.1932, 2010.

Rivière, G., and Orlanski, I.: Characteristics of the Atlantic storm-track eddy activity and its relation with the North Atlantic Oscillation, J. Atmos. Sc., 64, 241-266, doi: 10.1175/JAS3850.1, 2007.

Thurnherr, I., Kozachek, A., Graf, P., Weng, Y., Bolshiyanov, D., Landwehr, S., Pfahl, S., Schmale, J., Sodemann, H., Steen-Larsen, H. C., Toffoli, A., Wernli, H., and Aemisegger, F.: Meridional and vertical variations of the water vapour isotopic composition in the marine boundary layer over the Atlantic and Southern Ocean, Atmos. Chem. Phys., 20, 5811–5835, https://doi.org/10.5194/acp-20-5811-2020, 2020a.

Thurnherr, I., Hartmuth, K., Jansing, L., Gehring, J., Boettcher, M., Gorodetskaya, I., Werner, M., Wernli, H., and Aemisegger, F.: The role of air–sea fluxes for the water vapour isotope signals in the cold and warm sectors of extratropical cyclones over the Southern Ocean, Weather Clim. Dynam. Discuss. [preprint], https://doi.org/10.5194/wcd-2020-46, revised, 2020b.

Trenberth, K. E.: Atmospheric moisture residence times and cycling: Implications for rainfall rates and climate change, Clim. Change, 39, 667– 694, doi:10.1023/A:1005319109110, 1998.

Winschall, A., Pfahl, S., Sodemann, H., and Wernli, H.: Comparison of Eulerian and Lagrangian moisture source diagnostics – the flood event in eastern Europe in May 2010, Atmos. Chem. Phys., 14, 6605–6619, https://doi.org/10.5194/acp-14-6605-2014, 2014a.

Winschall, A., Sodemann, H., Pfahl, S., and Wernli, H.: How important is intensified evaporation for Mediterranean precipitation extremes?, J. Geophys. Res. Atmos., 119, 5240- 5256, doi:10.1002/2013JD021175, 2014b.